# Synthetic Data Generation of Many-to-Many Datasets via Random Graph Generation

**Kai Xu**[*]
Hazy
me@xuk.ai

**Georgi Ganev**[†]
Hazy
georgi@hazy.com

**Emile Joubert**
Hazy
emile@hazy.com

**Rees Davison**
Hazy
rees@hazy.com

**Olivier Van Acker**
Hazy
ovanac01@mail.bbk.ac.uk

**Luke Robinson**
Hazy
luke@hazy.com

## Abstract

Synthetic data generation (SDG) has become a popular approach to release private datasets. In SDG, a generative model is fitted on the private real data, and samples drawn from the model are released as the protected synthetic data. While real-world datasets usually consist of multiple tables with potential *many-to-many* relationships (i.e. *many-to-many datasets*), recent research in SDG mostly focuses on modeling tables *independently* or only considers generating datasets with special cases of many-to-many relationships such as *one-to-many*. In this paper, we first study challenges of building faithful generative models for many-to-many datasets, identifying limitations of existing methods. We then present a novel factorization for many-to-many generative models, which leads to a scalable generation framework by combining recent results from random graph theory and representation learning. Finally, we extend the framework to establish the notion of $(\epsilon, \delta)$-differential privacy. Through a real-world dataset, we demonstrate that our method can generate synthetic datasets while preserving information within and across tables better than its closest competitor.

## 1 Introduction

Private data release has gained much attention in recent years due to new regulations in privacy such as General Data Protection Regulation (GDPR). To obey such regulations and to protect privacy, a popular approach from the machine learning community called *synthetic data generation* (SDG) is developed in various domains (Nowok et al., 2016; Montanez et al., 2018; Xu & Veeramachaneni, 2018; Xu et al., 2019; Lin et al., 2020; Tucker et al., 2020; Xu et al., 2021; Ziller et al., 2021). In SDG, synthetic data that is statistically similar but not identical to real data is released as a replacement of the real data to protect. On top of it, special attention has also been paid to make sure these methods work well on tabular data as most real-world datasets are stored as tables in databases (Montanez et al., 2018; Xu & Veeramachaneni, 2018; Xu et al., 2019; Nazabal et al., 2020; Ma et al., 2020).

At the core of most SDG methods are *generative models*—probabilistic models that one can drawn samples from. Usually a generative model is firstly trained to capture the distribution of the private real data, and then samples are drawn from the model so they can be released as protected synthetic data. However, this procedure without extra care does not have guarantees privacy. To tackle this, research in this direction has also been focusing on *differentially private generative models* (Zhang et al., 2017; Xie et al., 2018; Jordon et al., 2018)—generative models that satisfy a notion of privacy called *differential privacy* (Dwork et al., 2014), which controls the amount of information individual datum can reveal. Appendix A provides an illustration of how generative models are used for SDG.

While real-world datasets usually consist of multiple tables with potential many-to-many relationships, SDG for such type of data is not well-studied. Most of recent work focuses on modeling tables *independently* (Xu & Veeramachaneni, 2018; Xu et al., 2019; Nazabal et al., 2020; Ma et al., 2020) or considers generating many-to-many relationships but only for special cases such as *one-to-many* (Getoor et al., 2007; Montanez et al., 2018). As a side effect, privacy for multi-table SDG with many-

---

[*]Now at Amazon; work done prior to joining Amazon
[†]Also a Ph.D candidate at the University College London

(a) Bipartite graph $\mathcal{B}$ with annotations for node degrees

(b) Adjacency matrix $\mathbf{G}$ (left) and biadjacency matrix $\mathbf{B}$ (right)

(c) BJDD matrix $\mathbf{P_J}$ with supports $\mathbf{d}_u = \deg(n_u), \mathbf{d}_v = \deg(n_v)$

Figure 1: An illustration of bipartite graphs and their related representations and statistics.

to-many relationships is also under-studied. In fact, relationships in real data can also reveal private information (Sala et al., 2011; Proserpio et al., 2012). For example, in a customer-merchant dataset, the number of links to a merchant could reveal its identity because of its uncommon popularity.

This paper studies the challenge of building faithful[1] generative models (Webb et al., 2018) to synthesize data together with their many-to-many relationships, and proposes a novel factorization for many-to-many generative models, which leads to a scalable approach by combining results from random graph theory and representation learning. In short, our paper has the following contributions:

1. We study possible factorization of faithful generative models for many-to-many data and identify limitations of those taken by existing approaches.
2. We propose a novel factorization for modeling distributions over many-to-many data and we use it to develop a synthetic data generation framework using methods from random graph generation, node representation learning and set representation learning.
3. We extend the proposed framework to establish the notion of $(\epsilon, \delta)$-differential privacy.
4. We evaluate two model instances from our framework, BayesM2M and NeuralM2M, on the MOVIELENS dataset, demonstrating its superior performance over its closest competitor, SDV, especially on capturing information in many-to-many relationships.

## 2 BACKGROUND AND NOTATIONS

**Bipartite and multipartite graphs**    A *bipartite graph* $\mathcal{B} := (\mathbb{U}, \mathbb{V}, \mathbb{L})$ is a tuple of two disjoint node sets $\mathbb{U}, \mathbb{V}$ (called *upper* and *lower* nodes respectively) as well as a set of edges $\mathbb{L}$ between them. Each node is represented by a tuple of its index and attribute, e.g. $n_u^k := (i_u^k, x_u^k) \in \mathbb{U}$ for $k = 1, \ldots, |\mathbb{U}|$. Each edge is represented by a tuple of node indices, i.e. $l^k := (i_u^k, i_v^k) \in \mathbb{L}$ for $k = 1, \ldots, |\mathbb{L}|$. Such edge sets can be represented as *adjacency matrices* $\mathbf{G}$ or *biadjacency matrices* $\mathbf{B}$; see figure 1b for an illustration. A generalised notion of bipartite graphs called *multipartite graphs*, denoted as $\mathcal{M} := (\{\mathbb{T}^{k_1}\}_{k_1=1}^N, \{\mathbb{L}^{k_2}\}_{k_2=1}^M)$, are graphs with $N$ disjoint node sets and $M \leq \binom{N}{2}$ edge sets.

**Modeling datasets with many-to-many relationships**    [2]Datasets with many-to-many relationships, such as relational databases, can be viewed as multipartite graphs. Denote the generating distribution of a given multipartite graph $\mathcal{M}_{\text{data}}$ as $p_{\text{data}}(\mathcal{M})$. The goal of this paper is to build a generative model $p_\theta(\mathcal{M})$ with learnable parameter $\theta$ and to develop a learning algorithm $\mathcal{A}_1$ that takes $\mathcal{M}_{\text{data}}$ as inputs and outputs the optimal model $p_{\theta*}$ with parameter $\theta^*$, s.t. $p_{\theta*}(\mathcal{M})$ is close to $p_{\text{data}}(\mathcal{M})$. After learning, a sample $\tilde{\mathcal{M}}$ is drawn from the model and used as synthetic data; we denote this sampling process/algorithm as $\mathcal{A}_2$.

**Differential privacy**    In this work, we are interested in establishing the standard $(\epsilon, \delta)$-*differential privacy*, a.k.a. $(\epsilon, \delta)$-DP or approximate DP. Denote $\mathcal{A} = \mathcal{A}_2 \circ \mathcal{A}_1$. We say a model $p_\theta$ with $\mathcal{A}$ is $(\epsilon, \delta)$-DP according to the following definition (Dwork et al., 2014):

**Definition 1** $((\epsilon, \delta)$-DP for $p_\theta$ with $\mathcal{A})$. *A generative model $p_\theta$ with the synthesis algorithm $\mathcal{A}$ is $(\epsilon, \delta)$-DP if for all $\mathcal{S} \subseteq \text{Range}(\mathcal{A})$ and for all $\mathcal{B}, \mathcal{B}'$ that differ on a single element:*

$$\mathbb{P}(\mathcal{A}(\mathcal{B}) \in \mathcal{S}) \leq \exp(\epsilon)\mathbb{P}(\mathcal{A}(\mathcal{B}') \in \mathcal{S}) + \delta.$$

---

[1]The term "faithful" as in faithful generative models refers to the fact that the model does not introduce any conditional independence that is not true in general (Webb et al., 2018). It is true for how our proposed model factorizes individual tables as well as their relationships.

[2]Note here we are intended to avoid using terms such as primary/foreign keys and parents/children from the relational database literature. This is because in order to model the joint distribution for a given dataset, parent-child directions can be rearranged for modeling convenience in favor of some particular factorization of the joint.

In our case, $\mathcal{S}$ is the output of the synthesis algorithm $\mathcal{A}$'s training procedure, $\mathcal{B}$ is a many-to-many dataset while we are interested in the most general case (with strongest privacy protection) and define an element as a single node that could be part of either $\mathbb{U}$ or $\mathbb{V}$. If $\delta = 0$, we also call a model $p_\theta$ with $\mathcal{A}$ satisfies $\epsilon$-differential privacy, i.e. $\epsilon$-DP.

**Learning graph distributions**  Consider the distribution $p_{\text{data}}(\mathbb{L})$ for a given set of edges $\mathbb{L}_{\text{data}}$. Unlike setup in work like Liao et al. (2019) in which multiple samples from $p_{\text{data}}$ are given, we are presented with a single sample from $p_{\text{data}}$. Therefore, rather than replying on the standard notion of *learning as probability divergence minimization* using empirical samples, we consider learning a model $p_\theta(\mathbb{L})$ such that samples from it share similar *graph properties* as $\mathbb{L}_{\text{data}}$.

**Graph properties**  We consider the $dK$-*distributions* from Mahadevan et al. (2006) with $d = 2$, defined as the (joint) degree distribution of all pairs of $d$ nodes in the graph.[3] We call all graphs with a given $dK$-distributions $dK$-*graphs*. For example, $0K$-graphs share the same graph density, $1K$-graphs share the same distribution of node degrees, $2K$-graphs share the same joint distributions of degrees of node pairs, etc. We call models that sample $dK$-graphs $dK$-*generators*. We consider the same notion for bipartite graphs, defining *bipartite $dK$-graphs*, and consider learning $p_\theta$ that generates bipartite $2K$-graphs of data. For a bipartite graph $\mathcal{B}$, we denote its *bipartite joint degree distribution* (BJDD) as the empirical distribution of samples $\{(\deg(\text{n}(i_u^k)), \deg(\text{n}(i_v^k)))\}_{k=1}^{|\mathbb{L}|}$, where $\text{n}(i)$ denotes the node with index $i$ (similarly $\text{i}(n)$ is the node index for node $n$) and $\deg(n)$ is the degree of the node $n$. Such empirical distribution can be represented as a matrix $\mathbf{P_J}$ along with the supports $\mathbf{d}_u, \mathbf{d}_v$ for both dimensions; see figure 1c for an illustration. One can also compute the (marginal) degree distributions of the upper or lower nodes from BJDD. For example, that of upper nodes $\mathbf{p}_u$ is a probability vector (a vector of whose elements sum up to 1) in which its $i$-th element $\mathbf{p}_u[i] = \sum_j \mathbf{P_J}[i,j]/\mathbf{d}_u[i]$.

## 3  RELATED WORK

**Generative models for tabular data**  In general, there are two categories of generative models that have been successfully applied to tabular data. One is based on classic *graphical models*. This includes *directed* graphical models such as Bayesian networks PrivBayes (PrivBayes; Zhang et al., 2017) and auto-regressive models (synthpop; Nowok et al., 2016) as well as *undirected* graphical models such as factor graphs (MST; McKenna et al., 2019). For these models, data is usually discretized in order to facilitate efficient learning. Another is based on recently emerged *deep generative models* such as *variational autoencoders* (VAEs; Kingma & Welling, 2013; Rezende et al., 2014) and *generative adversarial networks* (GANs; Goodfellow et al., 2014). Unlike graphical models, work on deep generative models has been mostly focusing on developing methods to directly model *heterogeneous data*, which are mixed-type data with different types of continuous (real, positive real, etc) and discrete variables (categorical, ordinal, etc), which includes specific pre-processing steps, neural network architecture design, improving training, etc (Xu & Veeramachaneni, 2018; Xu et al., 2019; Nazabal et al., 2020; Ma et al., 2020); for surveys, see Fan et al. (2020); Borisov et al. (2021).

**Generative models for relational data**  Probabilistic relational models (PRMs) (Friedman et al., 1999; Getoor et al., 2007) is the most well-established approach to building distributions over relational data. Getoor et al. (2007) describes two ways of dealing with structural uncertainty but only with learning methods for structures, i.e. finding the best structures. While it is possible to extend these methods to samplers by using Markov chain Monte Carlo for sampling, applying them to datasets at scale would be challenging. Mostly related to our work is *Synthetic Data Vault* (SDV) (Patki et al., 2016; Montanez et al., 2018), which is based on PRMs and considers the task of synthesizing relational data with links. However, SDV only models one-to-many relationships by recording the children count per row and generates one-to-many links by sampling the number of children. Because SDV cannot handle many-to-one relationships, the so-called "multiple parents" problem occurs in certain scenarios (Montanez et al., 2018). For example, consider three tables

---

[3]With $d$ increasing, samples from the $dK$-distributions become more similar to the data but generating $dK$-graphs is increasingly computationally complex; luckily $2K$-graphs are sufficient for most practical purposes (Mahadevan et al., 2006). Also note that with $d \to \infty$, $dK$-series describe any given graph completely. In addition, from a privacy perspective, a large $d$ means high sensitivity in the $dK$-series, which requires much more noise to be added for achieving $\epsilon$-DP for a given $\epsilon$ (Sala et al., 2011).

$A - B - C$ forming an *one-many-one* relationship. At generation time, SDV has to generate $A$ and $C$ first and conditionally generates $B$ given $A$ and $C$, at which stage it is unclear whether the children counts in $A$ or $C$ should be used. Montanez et al. (2018) suggests to either randomly pick a table for the conditional generation or to perform weighted averaging between parents. Another related work based on PRMs is Ben Ishak et al. (2016), which describes a generator for many-to-many data with links. However, the work only targets generating testing data—data used as test examples in a software engineering setup. Thus, the generator is learnable.

**Random graph generation** One line of work in this direction is built on the classic *degree sequence problem*, which is the problem of finding some or all graphs with a given degree sequence (Molloy & Reed, 1995). Mahadevan et al. (2006) introduces the notion of $dK$-graphs and studies generating them via classic random graph methods, such as the *pseudograph* or *configuration* approach (Molloy & Reed, 1995; Aiello et al., 2000). As classic methods in general cannot correctly sample $dK$-graphs with $d \geq 2$, a few follow-up works improves the sampling for these graphs (Stanton & Pinar, 2012; Gjoka et al., 2013; 2015). Unlike previous work that focuses on general graphs, Boroojeni et al. (2017) specifically considers sampling bipartite graphs that preserves the $2K$-distribution. Another line of methods, unlike above-mentioned methods that can generate arbitrary-sized graphs, can only generate graphs with the same size as the data graph, e.g. spectral-based approach in Baldesi et al. (2018), or expands it by a whole number, e.g. method based on fractal expansions from Belletti et al. (2019).

**Differentially private SDG** In terms of DP generative models, a few graphical models discussed above (PrivBayes, MST) are originally proposed in a DP context. For deep generative models, there are two generic training algorithms that make models differentially private: differentially private stochastic gradient descent (DP-SGD; Abadi et al., 2016) and private aggregation of teacher ensembles (PATE; Papernot et al., 2016; 2018), leading to methods such as DP-GAN (Xie et al., 2018) and PATE-GAN (Jordon et al., 2018). Regarding random graph generation, Sala et al. (2011); Proserpio et al. (2012) consider the release of graphs in a differentially private manner based on $dK$-generators. To the best of our knowledge, there is no work specifically targeting differentially private synthesis of bipartite graphs, nor previous work for differentially private synthesis for many-to-many datasets.

## 4 MODELING MANY-TO-MANY DATASETS WITH DIFFERENTIAL PRIVACY

In this section, we first discuss the challenges of building faithful generative models (Webb et al., 2018) for many-to-many datasets by studying possible factorization of the joint distribution (section 4.1), during which we also identify the limitations of the factorization that existing approaches are based on. Based on a novel factorization, we use methods from random graph generation, node representation learning and set representation learning to develop a new type of scalable generative model for many-to-many datasets section 4.2. Finally, we extend the model to be $(\epsilon, \delta)$-differentially private section 4.3. Also note that in this section, for brevity, we omit the model parameter $\theta$ in $p_\theta$ unless it is necessary.

### 4.1 FAITHFUL GENERATIVE MODELS FOR MANY-TO-MANY DATASETS

To simplify the discussion, we first focus on modeling bipartite graph and will later extend it to multipartite graphs. For the joint distribution $p(\mathbb{U}, \mathbb{V}, \mathbb{L})$, where $\mathbb{U}, \mathbb{V}$ are two tables with rows linked by edges in $\mathbb{L}$, that we are interested in, considering the fact that the roles of $\mathbb{U}$ and $\mathbb{V}$ are exchangeable, there are only three possible factorization of the joint $p(\mathbb{U}, \mathbb{V}, \mathbb{L}) := p_{\text{joint}}$:

| | FACTORIZATION S | FACTORIZATION E | FACTORIZATION A |
|---|---|---|---|
| $p_{\text{joint}}$ | $p(\mathbb{U})p(\mathbb{L} \mid \mathbb{U})p(\mathbb{V} \mid \mathbb{U}, \mathbb{L})$ | $p(\mathbb{U})p(\mathbb{V} \mid \mathbb{U})p(\mathbb{L} \mid \mathbb{U}, \mathbb{V})$ | $p(\mathbb{L})p(\mathbb{U} \mid \mathbb{L})p(\mathbb{V} \mid \mathbb{U}, \mathbb{L})$ |

Within all factorization, there are a 6 sub-models in total to consider.

- $p(\mathbb{U})$ (the single-table model) can be any generative model for single table generation.
- $p(\mathbb{L} \mid \mathbb{U})$ (the semi-conditional edge model) requires to generate edges based on one of the table—the row count generation used in Montanez et al. (2018) is in fact an example of such. However, a general case of this model can be computational challenging: For any subset of $\mathbb{U}$, it requires checking if they together link to an existing or new node in $\mathbb{V}$.
- $p(\mathbb{V} \mid \mathbb{U}, \mathbb{L})$ (the conditional table model) requires to generate each node in $\mathbb{V}$ based on the first table and all connections. As per node in $\mathbb{V}$ is connected to a subset of nodes in $\mathbb{U}$ via $\mathbb{L}$,

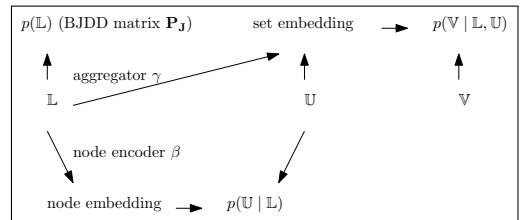 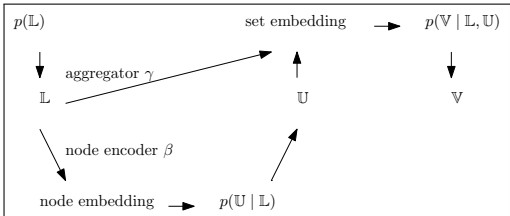

| (a) Learning (models can be fitted in parallel) | (b) Sampling (generation is sequential) |

Figure 2: Illustrations of how the proposed framework works during learning and sampling phases.

it requires to condition on a set of varied number of nodes for modelling, which can be done using a *aggregation function* or a *aggregator* (Getoor et al., 2007; Montanez et al., 2018).

- $p(\mathbb{V} \mid \mathbb{U})$ (the table-conditional table model) requires to generate one table conditioned on another (while marginalizing all edges). This model can be computational challenge to implement. A potential work-around is to model the two-table joint $p(\mathbb{U}, \mathbb{V})$ directly by joining two tables and apply a generative model on the joined table. However, this could easily lead to table with an intractable scale. And perhaps more importantly, it is hard to undo this join operation from the synthetic data to recover individual tables in general.[4].
- $p(\mathbb{L} \mid \mathbb{U}, \mathbb{V})$ (the edge prediction model) predicts the existences of edges between node pairs and is essential probabilistic models used in many recommender systems.
- $p(\mathbb{L})$ (the edge model) requires to model the edges (i.e. the graph) unconditionally—this is the type of random graph models that usually studied in random graph theory.
- $p(\mathbb{U} \mid \mathbb{L})$ (the edge-conditional table model) requires to generate one of the table given the topology of edges. One way to achieve such conditioning is by using a node embedding to condition on. This is also related to node attributes generation/prediction in graph models.

As discussed, approaches from Getoor et al. (2007); Montanez et al. (2018) correspond to FACTORIZA-TION S and there is no easy way to extend $p(\mathbb{L} \mid \mathbb{U})$ to a general many-to-many case. FACTORIZATION E also has the scalability issue in $p(\mathbb{V} \mid \mathbb{U})$ (or similarly in $p(\mathbb{U}, \mathbb{V})$) as discussed. Our approach will then be based on FACTORIZATION A that does not have such concerns/limitations.

**Considering edges attributes** So far our discussion assumes there are no edge attributes, which is not true in general, e.g. ratings in MOVIELENS. The way to deal with this is by adding an extra edge attribute model that is similar to $p(\mathbb{L} \mid \mathbb{U}, \mathbb{V})$ but generates edge attributes instead. One can use any edge prediction model for this purpose—rather than predicting the existence of edges it generates the edge attributes—and the corresponding generation step is performed after all other data is generated.

**Extending to multipartite graphs** For a given datasets with multiple tables, we assume the links at table level are given. Thus we can impose an *arbitrary* order on the tables and generate the edges and tables in order; this is similar to how an arbitrary order is imposed on random variables as in Bayesian networks. Extra care needs to be taken for graph generation when generating graphs involving tables with more than one connection, which will be explained in section 4.2.1.

## 4.2 RANDOM GRAPH GENERATION BASED MANY-TO-MANY SYNTHESIS

We now describe how to model the three distributions in our proposed factorization $p_{\text{joint}} = p(\mathbb{L})p(\mathbb{U} \mid \mathbb{L})p(\mathbb{V} \mid \mathbb{U}, \mathbb{L})$: $p(\mathbb{L})$ in section 4.2.1, $p(\mathbb{U} \mid \mathbb{L})$ in section 4.2.2 and $p(\mathbb{V} \mid \mathbb{L}, \mathbb{U})$ in section 4.2.3. Figure 2 also provides an illustration of the learning and sampling process for the proposed method.

### 4.2.1 GRAPH MODELING VIA RANDOM GRAPH GENERATION

We consider using bipartite $dK$-generators to model $p(\mathbb{L})$ that take in a target BJD **J** and generate random graph satisfying this **J**. Boroojeni et al. (2017) proves the existence of an algorithm for

---

[4]As a concrete example, suppose $\mathbb{U}$ has a single column $C_1$ with a continuous variable, $\mathbb{V}$ has another column $C_2$ (with either a continuous or discrete variable) and $\mathbb{L}$ indicates every row in $\mathbb{U}$ is linked with two rows in $\mathbb{V}$. After joining two tables, we will arrive with a table in which two columns $C_1, C_2$ in which there are duplicated elements in $C_1$ due to the one-to-many relationship; it is also possible to undo the join operation on this real dataset. However, for any generative model that treats $C_1$ as continuous variable, the probability of having duplicated elements in $C_1$ is exactly 0, which means any synthetic dataset would effectively giving a dataset with one-to-one relationship between $\mathbb{U}$ and $\mathbb{V}$, after undoing the join operation.

sampling bipartite graphs with a target BJD $\mathbf{J}$, giving rise to bipartite $2K$-generators. The idea is to first generate a bipartite graph based on random pairing, after which the graph matches the $2K$ distribution but has potentially repeated edges. Then, a rewiring process is performed to make the graph *simple*—a graph is called simple if there is no loop or multiple edges (West et al., 2001). Boroojeni et al. (2017) shows that such rewiring process exists and can always produce a graph with the target BJD. Since the prescribed algorithm of rewiring in Boroojeni et al. (2017) contains a few mistakes/typos, we present a corrected version in algorithm 1 in appendix B.

To complete the description of $p_\theta(\mathbb{L})$, we also need to define how to obtain $\mathbf{J}$ that bipartite algorithm 1 take in. For this, we draw samples from $\mathbf{P_J}$ with a prescribed number of edges. However, as this might invalidate the input to algorithm 1[5], which may cause infinite loops in rewiring, we stop the rewiring process after a predefined maximum number of iterations. To this end, sampling from $\mathbf{J}$ together with algorithm 1 define $p_\theta(\mathbb{L})$ through its generative process, where $\theta$ is the BJDD matrix $\mathbf{P_J}$.

**"Backward compatibility" of the graph generator**  One consideration taken into account while selecting the model for $p(\mathbb{L})$ was "backward compatibility" to one-to-many and many-to-one problems. This refers to the cases where if the data graph contains only one-to-many relationships, the generated graphs shall do so. The $2K$-generator indeed satisfies this property while, in contrast, spectral-based graph generation methods such as Baldesi et al. (2018) usually fail to hold it.

**Learning and sampling**  Learning of this model only involves computing the BJDD matrix from the data graph; if a table is involved in multiple edges, the joint distribution of all degrees per node is learned. Sampling from the model is defined by sampling from $\mathbf{P_J}$ followed by algorithm 1.

### 4.2.2 CONDITIONAL MODELING VIA NODE EMBEDDING

We now discuss how to model upper nodes $\mathbb{U}$ conditioned on edges $\mathbb{L}$. This distribution essentially models a set of nodes given topology, which can be dealt with the use of node embedding as:

$$p(\mathbb{U} \mid \mathbb{L}) = \prod_{k=1}^{|\mathbb{U}|} p(n_u^i \mid \beta(\mathbb{L}, i)), \tag{1}$$

where the node encoder $\beta$ computes *node embedding* of node $i$ in graph $\mathbb{L}$; its *range* is $\operatorname{ran}(\beta) = \mathbb{R}^{H_\beta}$, where $H_\beta$ is the dimension of the node embedding. Intuitively speaking, $\beta$ computes the representation of the local topology of node $i$ in $\mathbb{L}$; if the representation is rich enough, $n_u^i$ and $n_u^j$ are conditional independent given the representation, for any $i, j$. As such, once given $\beta$, this distribution can be modeled by any existing conditional generative model. A few candidates of such $\beta$ are:

1. Node statistics: $d$-step neighbour count (number of neighbours of distance $d$) can be used. As an simple example, $d = 1$ means using node degrees as 1d embeddings. It can be generalised to use a collection of counts with $H_\beta$ different $d$, giving to an embedding of size $H_\beta$.
2. Node embedding: Methods to learn node embedding such as DeepWalk (Perozzi et al., 2014) and node2vec (Grover & Leskovec, 2016) can also be used as $\beta$.

Addition discussion on independence assumptions behind equation 1 can be found in appendix C.

**Learning**  Learning of this model only involves learning the conditional generative model. There is no learning for either node statistics or node embedding (for the two approaches mentioned above).

### 4.2.3 CONDITIONAL MODELING VIA SET EMBEDDING

Consider a node $n_v \in \mathbb{V}$ with neighbors $\operatorname{ne}(n_v) \subseteq \mathbb{U}$. It is easy to see that, given $\operatorname{ne}(n_v)$, $n_v$ is conditionally independent with any other $n'_v \in \mathbb{V}$. Thus, we have $p(\mathbb{V} \mid \mathbb{L}, \mathbb{U}) = \prod_{n_v \in \mathbb{V}} p(n_v \mid \operatorname{ne}(n_v))$, where $p(n_v \mid \operatorname{ne}(n_v))$ is a conditional distribution whose condition is a *multiset*—a set that allows repeated elements. Without imposing any order on $\operatorname{ne}(n_v)$, one way to condition on this through a *fixed-size set embedding* computed by an aggregator $\gamma$, which leads to our model

$$p(\mathbb{V} \mid \mathbb{L}, \mathbb{U}) = \prod_{n_v \in \mathbb{V}} p(n_v \mid \gamma(\operatorname{ne}(n_v))). \tag{2}$$

---

[5]Algorithm 1 assumes two conditions for the input $\mathbf{J}$: (i) $\mathbf{J}[i, j] \leq (\sum_j \mathbf{J}[i, j]/\mathbf{d}_u[i])(\sum_i \mathbf{J}[i, j]/\mathbf{d}_v[j])$ and (ii) the degree sums for upper and lower nodes are the same (which would be true as long as $\sum_j \mathbf{J}[i, j]/\mathbf{d}_u[i]$ is integer for all $i$ and $\sum_i \mathbf{J}[i, j]/\mathbf{d}_u[j]$ is integer for all $j$) (Boroojeni et al., 2017).

Equation 2 requires an aggregator $\gamma : \mathrm{pow}(\mathbb{U}) \mapsto \mathbb{R}^{H_\gamma}$ where $\mathrm{pow}(\mathbb{U})$ is the *power set* of $\mathbb{U}$ and $H_\gamma$ is the dimension of the set embedding. If the set embedding of neighbours from $\gamma$ is rich enough, any two nodes $n_v$ and $n'_v$ are conditionally independent given the corresponding set embeddings. A few candidates of such $\gamma$ are:

1. Summary statistics: count, sum, $n$-th order moments, $q$-percent quantiles, etc. can be used. If both count and sum are used and $d$ different moments and quantiles are used, the total dimension of the embedding is $D + D + d \times D + d \times D$ for nodes with $D$-dimensional attributes.
2. Distribution parameters: One can also fit a distribution on the set and use the distribution parameters as the embedding. Montanez et al. (2018) suggests using a Gaussian copula; for nodes with $D$-dimensional attributes, the dimension of the embedding is $D + D^2$.
3. Set embedding: The deep sets architecture (Zaheer et al., 2017) is specifically proposed to produce a set embedding. It contains two neural networks, one to compute an embedding for each element in the set and one to aggregate the element embeddings into a set embedding.

Addition discussion on independence assumptions behind equation 2 can be found in appendix C.

**Learning**   Learning of this model involves learning the set embedding (if it is option 3) and learning the conditional generative model. Learning of the deep sets models can be difficult depends on the choice of the conditional generative model. However, if this model is also implemented by neural networks, one can learn $\gamma$ in an end-to-end fashion using the learning objective of the conditional generative model. As there is no existing method to train deep sets in an unsupervised manner, it cannot be used with classic models like Bayesian networks. We leave that to future work.

### 4.3   Establishing differential privacy

We now describe how to extend the learning of each sub-model in $p_\theta(\mathbb{U}, \mathbb{V}, \mathbb{L}) = p_{\theta_1}(\mathbb{L}) p_{\theta_2}(\mathbb{U} \mid \mathbb{L}) p_{\theta_3}(\mathbb{V} \mid \mathbb{L}, \mathbb{U})$ such that the learned parameter $\theta = (\theta_1, \theta_2, \theta_3)$ with corresponding privacy budgets $\epsilon_1$, $\epsilon_2$, and $\epsilon_3$ ($\epsilon_1 + \epsilon_2 + \epsilon_3 = \epsilon$) is $(\epsilon, \delta)$-DP, becomes $(\epsilon, \delta)$-DP by Theorem. 1 (defined below). Note that we assume the required privacy budgets for each component in each sub-model are given; how to optimally spend the budget is a question for future work that is out of the scope of this paper.

$p_{\theta_1}(\mathbb{L})$ **with** $\epsilon_1$   For $p_{\theta_1}$, $\theta_1$ is the BJDD matrix $\mathbf{P_J}$. As per the definition in section 2, we are interested in masking the presence of absence of a single node of either $\mathbb{U}$ or $\mathbb{V}$, which is a stronger privacy notion than edge protection. To do so we follow the strategy in Sala et al. (2011); Proserpio et al. (2012) that uses the Laplace mechanism to make $\mathbf{P_J}$ $\epsilon$-DP. The main difference here is that we are operating on bipartite graphs instead of general graphs. Specifically, after measuring the BJD matrix $\mathbf{J}$ from data graphs, to achieve $\epsilon$-DP, we add noise following $\mathcal{L}ap(\frac{S_{ij}}{\epsilon})$ where $S_{ij} = 4 \max(d_i, d_j)$ to the $i$-th row, $j$-th column entry of $\mathbf{J}$ (Sala et al., 2011; Proserpio et al., 2012).

$p_{\theta_2}(\mathbb{U} \mid \mathbb{L})$ **with** $\epsilon_2$   For $p_{\theta_2}$, $\theta_2$ consists the parameter of the underlying generative model for the conditional, for which we choose some existing DP ones and follow their learning algorithms.

$p_{\theta_3}(\mathbb{V} \mid \mathbb{L}, \mathbb{U})$ **with** $\epsilon_3$   For $p_{\theta_3}$, $\theta_3$ consists both the parameter of the set embedding model as well as that of the underlying generative model for the conditional. For the former, if such embedding is pre-fixed (referred as "statistics" in section 4.2.3) or from Gaussian copula (the second option in section 4.2.3), there is no learnable parameter for the node embedding model; if such embedding is based on the deep sets architecture, which is only defined when it can be trained end-to-end with the generative model, we use DP-SGD (Abadi et al., 2016) for training. For the underlying generative model not trained jointly, we choose some existing DP ones and follow their learning algorithms.

**Theorem 1** (($\epsilon, \delta$)-DP for $p_\theta$ with $\mathcal{A}$). *Given the factorization,* $p_\theta(\mathbb{U}, \mathbb{V}, \mathbb{L}) = p_{\theta_1}(\mathbb{L}) p_{\theta_2}(\mathbb{U} \mid \mathbb{L}) p_{\theta_3}(\mathbb{V} \mid \mathbb{L}, \mathbb{U})$, *as well as* $p_{\theta_1}(\mathbb{L})$ *being* ($\epsilon_1, \delta$)-DP, $p_{\theta_2}(\mathbb{U} \mid \mathbb{L})$ *being* ($\epsilon_2, \delta$)-DP, *and* $p_{\theta_3}(\mathbb{V} \mid \mathbb{L}, \mathbb{U})$ *being* ($\epsilon_3, \delta$)-DP, *by sequential composition it follows that* $p_\theta(\mathbb{U}, \mathbb{V}, \mathbb{L})$ *is* ($\epsilon_1 + \epsilon_2 + \epsilon_3, \delta$)-DP.

It should be noted that the proposed formulation is flexible enough to allow for the privacy protection of the nodes from only one of $\mathbb{U}$ and $\mathbb{V}$ but not the other. For instance, if we choose to protect only $\mathbb{U}$, we can set $\epsilon_3 = \infty$ when learning $p_{\theta_3}(\mathbb{V} \mid \mathbb{L}, \mathbb{U})$ and $d_j = 0$ for all $j$ when learning $p_{\theta_1}(\mathbb{L})$, making the overall synthetic data generation process $(\epsilon, \delta)$-DP ($\epsilon_1 + \epsilon_2 = \epsilon$).

# 5 EVALUATION

**Dataset**   We consider the MOVIELENS in our evaluation. MOVIELENS is a dataset that contains users' ratings to different movies (Harper & Konstan, 2015). The user and movie tables have their own attributes such as user ages and movie genres and have a many-to-many relationship through existing ratings. We choose MOVIELENS as it is a commonly used datasets to study graph properties (Harper & Konstan, 2015; Baldesi et al., 2018; Belletti et al., 2019).

**Pre-processing**   We discretize all features into maximally 30 bins. Note this discretization step is optional: for our study, it is meant to make modeling and evaluation easier, which is an orthogonal consideration to the main point of the study which is to assess the performance of many-to-many modelling. Also note we do not use any privacy budget during pre-processing by assuming all information needed are public, which is again a separate concern that is out of the scope of this paper.

## 5.1 QUALITY OF MANY-TO-MANY MODELING

In this section, we evaluate the quality of the proposed model in terms of synthetic data similarity. We consider a set of baselines as well as two realizations of the proposed method, which we explain next.

**Baseline**   We consider SDV (Montanez et al., 2018) as the baseline method in our evaluation.[6] SDV decomposes many-to-many relationships into multiple one-to-many ones and has the "multiple parents" (discussed in section 3). To handle this, instead of randomly picking one table as the parent as suggested by Montanez et al. (2018), we experiment both options and report the one with better results.

**Two model instances from the proposed framework**   The proposed framework enables a range of new generative models with different options in each submodel detailed in section 4.2. In this evaluation, we focus on two typical ones built on different single table models: a Bayesian network with the network building method from Zhang et al. (2017), referred as BN, and an autoregressive model in which each conditional is modelled by a neural network, referred as NeuralAR. The two realizations are then (i) a classic non-parametric approach in which the single table models used are BN, referred as Bayesian many-to-many (BayesM2M) and (ii) a neural-based approach in which the single table models used are NeuralAR, referred as neural many-to-many (NeuralM2M). In both realizations, for $p(\mathbb{L})$, we use the $2K$-generator. In BayesM2M, for $p(\mathbb{U} \mid \mathbb{L})$, we use node statistics with $d$-step counts for $d = 1, 2$; for $p(\mathbb{V} \mid \mathbb{L}, \mathbb{U})$, we use summary statistics with count, sum, $n$-th order moments for $n = 1, 2, 3$ and $q$-percent quantiles for $q = 0.25, 0.5, 0.75$. In NeuralM2M, for $p(\mathbb{U} \mid \mathbb{L})$, we use node2vec to extract node embeddings; for $p(\mathbb{V} \mid \mathbb{L}, \mathbb{U})$, we use the deep sets architecture (Zaheer et al., 2017) to consume a set of node (equivalently a set of rows).

**Metrics**   As the primary goal of SDG is to mimic the distribution of the real data, we consider three sets of metrics based on distributional similarity to evaluate the generation quality. The first set consists metrics that specifically evaluate the *quality of relationship generation*, which are one minus the total variation distances between the $dK$-distributions of real and synthetic data for $d = 1, 2$ (bounded between 0 and 1). The second set consists the pairwise mutual information (MI) similarity for each table to evaluate *the quality of individual tables*. For a pair of real and synthetic tables, we first compute the normalized MI[7] between each pair of columns in each table; for tables with $m$ features, this gives a pair of $m \times m$ matrices. The MI similarity is then defined as the average Jaccard indices between each pair of entries. Compared to alternative metrics such as $f$-divergences, this metric is bounded between 0 and 1 thus is easy to interpret. The third set consists the metrics to capture correlations between table pairs rooted in their many-to-many relationships, i.e. *cross-table similarity*. For each pair, we join the tables by their links into a single table and compute the pairwise MI similarity for each pair between two tables; for a pair of two tables with $m$ and $n$ features, this gives a pair of $m \times n$ matrices. The MI similarity is then defined as the average Jaccard indices between each pair of entries. To help understand our metrics, we provide detailed definitions with step-by-step examples in appendix D.

Table 1 summarizes the main results from experiments with the above setups. Compared to SDV, both

---

[6]We use the official implementation of SDV at `github.com/sdv-dev/SDV` in our experiments.

[7]For discrete $X, Y$, we have $I(X; Y) \leq \min(H(X), H(Y))$. Thus normalized MI is defined as $I(X; Y)/\min(H(X), H(Y))$, which is bounded between 0 and 1.

Table 1: Quality of many-to-many modelling on MOVIELENS. All metrics are bounded between 0 and 1 with 1 being the theoretical maximal value, i.e. higher the better. Numbers are mean and standard deviation based on 5 runs with different random seeds.

| Method | Degree Similarity | | Mutual Information Similarity | | |
| :---: | :---: | :---: | :---: | :---: | :---: |
| | Marginal | Joint | $\mathbb{U}$ | $\mathbb{V}$ | Cross-Table |
| SDV | $0.867 \pm 0.003$ | $0.614 \pm 0.012$ | $0.547 \pm 0.033$ | $0.334 \pm 0.017$ | $0.440 \pm 0.035$ |
| BayesM2M | $0.955 \pm 0.016$ | $0.634 \pm 0.028$ | $0.756 \pm 0.041$ | $0.345 \pm 0.016$ | $0.561 \pm 0.031$ |
| NeuralM2M | | | $0.846 \pm 0.045$ | $0.562 \pm 0.031$ | $0.613 \pm 0.016$ |

realizations of the proposed method obtains better scores in terms of graph similarity (marginal and joint degree similarity). This supports our choice of using random graph generation as the starting point of the whole generative process. Among two realizations, NeuralM2M has a better score in MI similarity for $\mathbb{U}$, indicating that the node representation from node2vec contains more information than a simple-minded node statistics. On top of this, NeuralM2M also has a larger the gain/different between MI similarity for $\mathbb{U}$ and $\mathbb{V}$ than BayesM2M, indicating that deep sets architecture learn better set representation than fixed summary statistics. As a result of using random graph generation and learnable neural-based representations, NeuralM2M has the best cross-table similarity. We present more results on further metrics in appendix D.

## 5.2 PERFORMANCE UNDER DIFFERENTIAL PRIVACY CONSTRAINTS

For the two typical realizations of the proposed framework, we now evaluate how the generation quality of their differentially private variants changes with different privacy budgets. We focus on the cross-table similarity in this section as it is the most comprehensive metric for many-to-many modelling.

**Privacy budgets** For both realizations, we vary $\epsilon$ in $\{0.1, 10.0, 1,000.0, \infty\}$. For NeuralM2M, we in addition set $\delta = 1 \times 10^{-5}$ and use a sensitivity (gradient clipping norm) of $5.0$ in DP-SGD.

Figure 3 summarizes the main results from experiments with the above setups. As it can be seen, even though NeuralM2M performs well in a non-DP setup, its performance drops quickly with even a slightly smaller privacy budget. In comparison, BayesM2M has a better privacy-similarity trade-off. The main reason behind this is that the components in NeuralM2M rely on iterative training in which noise is added per iteration in order to establish differential privacy, making it hard to optimally spend the privacy budget or to obtain a tight DP bound. On the other side, most of the component in BayesM2M are pre-fixed and BNs can achieve DP by only adding noises to the measurement (i.e. counting for discreized data) of each table column.

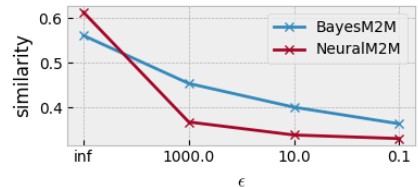

Figure 3: Cross-table similarity with privacy budgets $\epsilon$ on MOVIELENS

## 6 DISCUSSION AND CONCLUSION

**Limitation** The overall quality of the proposed framework depends on the choice of random graph generation methods. In this paper, we mostly focus on the $2K$-generator that can generate arbitary-sized graphs but it at its best matches the $2K$-distributions of the real data. Some other graph generation methods such as Baldesi et al. (2018); Belletti et al. (2019) can capture more graph statistics but can only generate graphs with the same size as the data graph or expands it by a whole number.

**Societal impacts** On the positive side, the proposed method improve SDG for many-to-many data and can be used to release such data with DP guarantees, protecting the privacy of relevant individuals. However, as all generative model-based SDG methods, the synthetic data may be different from the real one in certain ways, causing potential disparity issues.

To conclude, this paper studies the challenge of building faithful generative models for many-to-many datasets and present a novel generation framework based on random graph generation, node and set representation learning. The proposed framework is demonstrated to be better than its closest competitor SDV in terms of the generation quality on the widely-studied MOVIELENS dataset. We also study the behaviour of proposed model under different DP constraints, highlighting potential future works.

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

## A  HOW GENERATIVE MODELS ARE USED FOR SYNTHETIC DATA GENERATION

Figure 4 provides a flow chart to demonstrate the high-level workflow of generative models as applied to synthetic data generation in practice. As a more concrete example, suppose the real data is a many-to-many dataset that contains user reviews of various movies: It contains a table of users, a table of movies and a table describes how each user rates a movie (in a many-to-many fashion as a user could rate multiple movies and a movie can be reviewed by multiple users). Such a dataset may

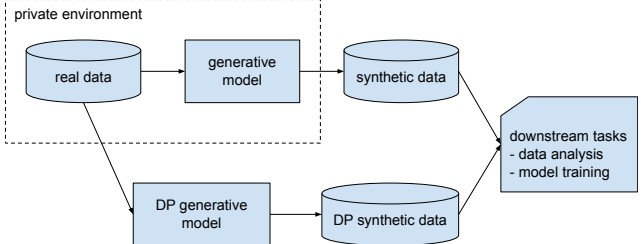

Figure 4: How generative models are used for synthetic data generation. To begin with, the real data stays in a private environment and cannot be shared with a third party due to privacy compliance. Depending on whether or not the synthetic data needs to be DP (according to compliance), the use of the generative model goes through a different route. If no DP is required, we can fit a generative model on the real data, both of which needs to stay in the private environment. Then synthetic data is sampled from the model, leaves the private environment and can be shared with a third-party to perform any downstream tasks of interest. If DP is needed for the synthetic data, we fit a DP generative model on the real data and the model can leave the private environment as the model is also DP. The rest of the process follows similarly to that of the non-DP route.

contain sensitive information such as user information (which may contain personally identifiable information) and how users rate movies (a user may be identified by how many movies he or she rated). The two routes of how generative models can be used to provide a synthetic copy of the data resolves this problem, especially the DP route. Downstream tasks such as data analysis (e.g. analysis of user behaviour) or model training (e.g. to train recommender systems) can then be performed on the synthetic data that is shared publicly by any third-party.

## B  BIPARTITE $2K$-GENERATORS

The complete algorithm of the bipartite $2K$-generators from Boroojeni et al. (2017) with corrections is given in algorithm 1. The algorithm consists two parts. The first part (Line 4 to Line 18) is called random pairing. It works by first creating stubs for each node based on its degree; the stubs for a node with index $i$ and degree $d$ is a list $\underbrace{[i, \ldots, i]}_{d}$. After stubs for all nodes are created, they are randomly paired based on the joint degree distribution/counts. This is simply done by enumerating the joint degree count: At each iteration for a pair of degree $d_1, d_2$, we randomly sample $d_1$ stubs with degree $d_1$ and $d_2$ stubs with degree $d_2$ and pair them accordingly. This step ensures the joint degree distribution matches while with *potential repeated edges*. The second part (Line 20 to Line 52) resolves repeated edges while preserving the joint degree counts. Boroojeni et al. (2017) shows that there are only two cases how such repeated edges could appear and how to resolve each of them accordingly with the joint degree counts unchanged. Note that the algorithm presented in Boroojeni et al. (2017) contains several mistakes even though their proof is correct; algorithm 1 is based on our own realization of situations in the proof but essentially the same algorithm.

## C  INDEPENDENCE ASSUMPTIONS

**Node embedding models**   As mentioned in section 4.2.2, equation 1 only holds if the representation from $\beta$ is rich enough such that any two nodes $n_u$ and $n'_u$ are conditional independent given the representations. This is more likely to be true if node embedding (option 2 above) is used. However, node statistics (option 1) is still appealing due to its simplicity and computational efficiency thus there exists an inherent fidelity-computation trade-off between these two modeling choices.

**Set embedding models**   As mentioned in section 4.2.3, equation 2 only holds if the embedding from $\gamma$ is rich enough such that any two nodes $n_v$ and $n'_v$ are conditionally independent given the corresponding set embeddings. This can be made true if set embedding (option 3 above) is used with a large enough dimension (Wagstaff et al., 2019) This could be potentially made true if distribution parameters (option 2 above) is used, if the distribution family is flexible enough. However, summary

# D   DETAILED EEXPLANATION OF METRICS USED IN SECTION 5

In section 5 we used a set of metrics that are not commonly used for evaluation. To repeat, the two reasons we prefer these metrics are (i) they are bounded between $0$ and $1$ and (ii) they are easy to compute once the data is discretized, e.g. not requirement for training a classifier for density ratio estimation in computing $f$-divergences. We now provide a more formal description of these metrics.

## D.1   DEGREE SIMILARITY

Let $\mathbf{p}, \mathbf{q}$ be the probability vectors representing the $1K$-distribution (with the same support) of the real and synthetic graphs. The *marginal degree similarity* is defined as

$$1 - \frac{1}{2}\text{TV}(\mathbf{p}, \mathbf{q}) = 1 - \frac{1}{2}\sum_i |p_i - q_i|$$

where TV is the total variation distance and $p_i, q_i$ are the $i$-th element of $\mathbf{p}, \mathbf{q}$.

Let $\mathbf{P}, \mathbf{Q}$ be the probability matrices representing the corresponding $2K$-distribution (with the same support). Similarly, the *joint degree similarity* is defined as

$$1 - \frac{1}{2}\text{TV}(\mathbf{P}, \mathbf{Q}) = 1 - \frac{1}{2}\sum_{i,j} |P_{ij} - Q_{ij}|$$

where $P_{ij}, Q_{ij}$ are the $i$-th row, $j$-th column entry of $\mathbf{P}, \mathbf{Q}$.

As $\text{TV}(\cdot, \cdot) \in [0, 2]$, it can be easily shown that the two above degree similarity metrics are bounded by $0$ and $1$ with $1$ being the maximal and best value.

## D.2   MUTUAL INFORMATION SIMILARITY

Let $\mathbf{X}, \mathbf{Y}$ be two datasets with the same number of discrete columns/features ($m$). For each dataset, we compute the normalized mutual information between each pair of columns, giving the pairwise MI matrix.

The normalized mutual information between two random variables $X, Y$ is defined as

$$\text{nMI}(X, Y) := \frac{\text{MI}(X, Y)}{\min\{\text{H}(X), \text{H}(Y)\}} = \frac{\text{H}(X) + \text{H}(Y) - \text{H}(X, Y)}{\min\{\text{H}(X), \text{H}(Y)\}}$$

where MI refers to the mutual information and H refers to the Shannon entropy. This quantity is bounded by $0$ and $1$ due to (i) the positivity of entropy and (ii) the inequality below

$$
\begin{aligned}
\text{H}(X, Y) &\geq \max\{\text{H}(X), \text{H}(Y)\} & \text{property of joint entropy} \\
-\text{H}(X, Y) &\leq \min\{-\text{H}(X), -\text{H}(Y)\} & \text{multiply by } -1 \\
\text{H}(X) + \text{H}(Y) - \text{H}(X, Y) &\leq \text{H}(X) + \text{H}(Y) + \min\{-\text{H}(X), -\text{H}(Y)\} & \text{add marginal entropies} \\
\text{MI}(X, Y) &\leq \min\{\text{H}(X), \text{H}(Y)\} & \text{simplification}
\end{aligned}
$$

Then the pairwise MI matrix, e.g. $\mathbf{M}$ for $\mathbf{X}$ can be defined by its $i$-th row, $j$-th column entries $M_{ij}$

$$M_{ij} := \text{nMI}(X_i, X_j)$$

where $X_i, X_j$ are the random variables for the $i$-th and $j$-th columns of the dataset $\mathbf{X}$. As the dataset is assumed to be given in discrete, the entropy of each column and the joint entropy of a pair of columns can be easily estimated by first estimating the empirical probabilities followed by the definition of Shannon entropy.

Let $\mathbf{M}, \mathbf{N}$ be the pairwise MI matrices for $\mathbf{X}, \mathbf{Y}$ respectively, the MI similarity is then defined as

$$\frac{1}{m^2}\sum_{i,j} \text{J}(M_{ij}, N_{ij})$$

Table 2: Quality of many-to-many modelling on MOVIELENS. All metrics are bounded between 0 and 1 with 1 being the theoretical maximal value, i.e. higher the better. Numbers are mean and standard deviation based on 5 runs with different random seeds.

| Method | Total Variation Similarity | | |
| --- | --- | --- | --- |
| | $\mathbb{U}$ | $\mathbb{V}$ | Cross-Table |
| SDV | $0.855 \pm 0.032$ | $0.937 \pm 0.001$ | $0.624 \pm 0.004$ |
| BayesM2M | $0.758 \pm 0.037$ | $0.951 \pm 0.027$ | $0.742 \pm 0.038$ |
| NeuralM2M | $0.800 \pm 0.061$ | $0.926 \pm 0.012$ | $0.745 \pm 0.046$ |

Table 3: Quality of many-to-many modelling on MOVIELENS. All metrics are bounded between 0 and 1 with 1 being the theoretical maximal value, i.e. higher the better.

| Recommender System | SDV | BayesM2M | NeuralM2M | Real Data |
| --- | --- | --- | --- | --- |
| F1 score | 0.522 | 0.507 | 0.591 | 0.591 |

where $M_{ij}, N_{ij}$ the $i$-th row, $j$-th column entries accordingly and J is the Jaccard index defined as

$$\mathrm{J}(A, B) = \frac{\min\{A, B\}}{\max\{A, B\}}.$$

It is easy to see this metric is bounded by 0 and 1 with 1 being the maximal and best value.

**Cross-table MI similarity**  To compute the cross-table MI similarity score, for a pair of real tables $X_1, X_2$ and synthetic tables $Y_1, Y_2$, we first join (join as in relational algebra or databases) the two table pairs ($X_1, X_2$ and $Y_1, Y_2$) into X and Y by the graph/links between them. Then, we simply follow the MI similarity described above to compute the score.

### D.3  TOTAL VARIATION SIMILARITY

Tao et al. (2021) uses a similar metric as in appendix D.2 where the pairwise mutual information between marginals is replaced by pairwise total variation distance (which is between 0 and 1, as defined by $\mathrm{TV}(P, Q) = \sum_{x \in \mathcal{X}} |P(\mathcal{X} = x) - Q(\mathcal{X} = x)|/2$ for two discrete distributions $P, Q$ over domain $\mathcal{X}$). We also report this metric with the only modification being that we use $1 - \mathrm{TV}(P, Q)$ instead, which makes the number higher the better for consistent interpretability as the mutual information similarity (while still being bounded between 0 and 1). We refer this metric as *total variation similarity*. Table 2 shows the results. Note that even the single-table performance, which is mainly dependent on the choice of single-table models, of ours is slightly worse than that of SDV (column $\mathbb{U}$) or similar (column $\mathbb{V}$), the cross-table performance of ours is still better with a noticeable margin (column Cross-Table). This is a strong indication of the benefit of our proposed many-to-many modeling approach.

### D.4  RECOMMENDER SYSTEM

Finally, we train a recommender systems on both real and synthetic datasets and compare their performance on a set aside test data. We report the results in Table 3. We see that both SDV and BayesM2M experience similar drop in performance compared to the recommender trained on the real data. For BayesM2M, we conjecture that the drop is because that the maximum number of parents is limited to 2, meaning no higher-level correlation is captured. Similar explanation can be made to the drop for SDV, where the single-table model (Gaussian copula) can only capture pairwise correlations. On the other hand, NeuralM2M achieves the same score as than the real due to the more powerful single-table model as well as our many-to-many modeling framework. This is perhaps surprising, as the the recommender system trained on the real data observes (almost) the same movies in the fitting and predicting steps while the NeuralM2M one was trained entirely on synthetic movies.

# E  REPRODUCIBILITY

## E.1  CODE

Our implementations of BayesM2M and NeuralM2M are available at `github.com/hazy/m2m`.

## E.2  EXPERIMENT DETAILS

In this section we provide some experimental details that are missing in section 5.

**Bayesian networks**  For BNs, we set the number of parents to be 2. The network building algorithm follows (Zhang et al., 2017) exactly.

**Neural autoregressive models**  For NeuralARs, the architecture for each classifier is as follows:[8]

1. Each discrete variable is passed to an embedding layer with an embedding size of 20, after which all embeddings are concatenated, giving a input embedding of size $d_e$
2. *(Optional)* When the deep sets architecture is used, each node is passed to a linear layer with output size 10 (i.e. node embedding size), then summed (i.e. summation as the aggregation function) and passed to a linear layer with output size 50 (i.e. set embedding size). We also overload the notation $d_e$ by this set embedding size if this step is performed.
3. The input embedding is passed to a multi-layer perceptron of $d_e \xrightarrow{\text{ReLU}} 50 \xrightarrow{\text{ReLU}} 50 \xrightarrow{\text{Softmax}} d_o$ where ReLU refers to the rectified linear unit activation function, Softmax refers to the he softmax function and $d_o$ is the output dimension, equal to the number of unique values of the output features.

Data is split by 80/20 into training and validation sets. All training is done using the ADAM optimizer (Kingma & Ba, 2014) for maximally $1,000$ steps with a learning rate of $1 \times 10^{-3}$ and a batch size of $500$. We also use a learning rate scheduler with the sine function with exponential amplitude decay with a minimal learning rate of $2 \times 10^{-4}$ and a 5-epoch periodicity.[9] Finally, early training is used based on the loss computed on the validation set: if there is no loss drop for 5 epochs, the training is early stopped.

**node2vec**  We use the Julia implementation of node2vec.[10] We keep all default parameters except setting the number of walks to be 10, the walk length to be 100, $p = q = 2$ and an embedding size of 20.

**Privacy accountant**  We apply a simple-minded way to spread the privacy budget. First, the privacy budget is evenly split among three sub-models. Second, within each sub-model, the budget is further evenly split to each component. For BNs, it means the network building and counting. For NeuralARs, we simply record the number of iterations used for training per classifier in the non-private version and evenly spread the privacy across all iterations. Note this (assuming the number of iterations per classifier is known) is for evaluation purpose only and by no means this strategy is an optimal way to spend the privacy budgets.

## E.3  TOTAL COMPUTATION RESOURCES

All experiments are conducted on an Amazon EC2 instance of type `c5.4xlarge` with CPUs only.

---

[8]All neural networks are implemented using Flux.jl (`https://github.com/FluxML/Flux.jl`.)

[9]This is done using the `SinExp` from ParameterSchedulers.jl (`https://github.com/darsnack/ParameterSchedulers.jl`).

[10]The code is available at `https://github.com/ollin18/Node2Vec.jl`.

---

**Algorithm 1** Bipartite $2K$-generator (Boroojeni et al., 2017, corrected)

---

1: **Input**: BJD $\mathbf{J}$ with supports $\mathbf{d}_u, \mathbf{d}_v$
2: **Output**: Edges $\mathbb{L}$
3: Initialise $\mathbb{L} = \emptyset$                                        ▷ Initialise edges as empty set
4: *# Part 1: random pairing*
5: $\mathbf{s}_u \leftarrow [], \mathbf{s}_v \leftarrow []$                         ▷ Initialise stubs as empty lists
6: Compute degree sequences $\mathbf{u}, \mathbf{v}$ from $\mathbf{J}$
7: **for** $i_u \in 1, \ldots, |\mathbf{u}|, k \in 1, \ldots, \mathbf{u}[i_u]$ **do**
8:     Append $i_u$ to $\mathbf{s}_u$
9: **end for**
10: **for** $i_v \in 1, \ldots, |\mathbf{v}|, k \in 1, \ldots, \mathbf{v}[i_v]$ **do**
11:     Append $i_v$ to $\mathbf{s}_v$
12: **end for**
13: **for** $i \in 1, \ldots, |\mathbf{d}_u|, j \in 1, \ldots, |\mathbf{d}_v|$ **do**
14:     $k \leftarrow \mathbf{J}[i, j]$
15:     Randomly pop $k$ node indices $\{i_u^l\}_{l=1}^k$ from $\mathbf{s}_u$ s.t. the corresponding nodes $n(i_u^l)$ has degree $\mathbf{d}_u[i]$
16:     Randomly pop $k$ node indices $\{i_v^l\}_{l=1}^k$ from $\mathbf{s}_v$ s.t. the corresponding nodes $n(i_v^l)$ has degree $\mathbf{d}_v[j]$
17:     Add edges $\{(i_u^l, i_v^l)\}_{l=1}^k$ to $\mathbb{L}$
18: **end for**
19: *# Part 2: rewiring process (corrected version)*
20: $\mathbb{L}_r \leftarrow$ repeated edges in $\mathbb{L}$                     ▷ Repeated edges to resolve
21: **while** $\mathbb{L}_r$ is not empty **do**
22:     Randomly pop edge $(i_u, i_v)$ from $\mathbb{L}_r$
23:     $n_u \leftarrow \mathrm{n}(i_u), n_v \leftarrow \mathrm{n}(i_v), n_u' \leftarrow n_u, n_v' \leftarrow n_v$
24:     $\mathbb{U}' \leftarrow \{n_u : \deg(n_u) = \deg(n_u)\} \setminus \{n_u\}$
25:     $\mathbb{V}' \leftarrow \{n_v : \deg(n_v) = \deg(n_u)\} \setminus \{n_u\}$
26:     **if** $|\mathbb{U}'| = 0$ **then**
27:         $n_v' \leftarrow$ uniform sample from $\{n_v' \in \mathbb{V}' \setminus \mathrm{ne}(n_u') : \mathrm{ne}(n_v') \setminus \mathrm{ne}(n_v) \neq \emptyset\}$
28:     **else if** $|\mathbb{V}'| = 0$ **then**
29:         $n_u' \leftarrow$ uniform sample from $\{n_u' \in \mathbb{U}' \setminus \mathrm{ne}(n_v') : \mathrm{ne}(n_u') \setminus \mathrm{ne}(n_u) \neq \emptyset\}$
30:     **else if** $\mathbb{L}_n' := \{(n_u', n_v') \in \mathbb{V}' \times \mathbb{U}' : \mathrm{ne}(n_v')) \setminus \mathrm{ne}(n_v) \neq \emptyset$ and $\mathrm{ne}(n_u') \setminus \mathrm{ne}(n_u) \neq \emptyset\} \neq \emptyset$
    **then**
31:         $(n_u', n_v') \leftarrow$ uniform sample from $\mathbb{L}_n'$
32:     **else if** $\mathbb{V}'' := \{n_v' \in \mathbb{V}' \setminus \mathrm{ne}(n_u')\} \neq \emptyset$ **then**
33:         $n_v' \leftarrow$ uniform sample from $\mathbb{V}''$
34:     **else** ▷ Boroojeni et al. (2017) proves that in this branch the set to draw samples is non-empty.
35:         $n_u' \leftarrow$ uniform sample from $\{n_u' \in \mathbb{U}' \setminus \mathrm{ne}(n_v')\}$
36:     **end if**
37:     **if** $n_u' = n_u$ **then**                                          ▷ Case 2
38:         $n_w' \leftarrow$ uniform sample from $\mathrm{ne}(n_v') \setminus \mathrm{ne}(n_v)$
39:         Remove edge $(\mathrm{i}(n_w'), \mathrm{i}(n_v'))$ from $\mathbb{L}$
40:         Add edges $\{(\mathrm{i}(n_u), \mathrm{i}(n_v')), (\mathrm{i}(n_w'), \mathrm{i}(n_v'))\}$ to $\mathbb{L}$
41:     **else if** $n_v' = n_v$ **then**                                     ▷ Case 2
42:         $n_w \leftarrow$ uniform sample from $\mathrm{ne}(n_u') \setminus \mathrm{ne}(n_u)$
43:         Remove edge $(\mathrm{i}(n_u'), \mathrm{i}(n_w))$ from $\mathbb{L}$
44:         Add edges $\{(\mathrm{i}(n_u'), \mathrm{i}(n_v)), (\mathrm{i}(n_u), \mathrm{i}(n_w))\}$ to $\mathbb{L}$
45:     **else**                                                             ▷ Case 1
46:         $n_w \leftarrow$ uniform sample from $\mathrm{ne}(n_u') \setminus \mathrm{ne}(n_u)$
47:         $n_w' \leftarrow$ uniform sample from $\mathrm{ne}(n_v') \setminus \mathrm{ne}(n_v)$
48:         Remove edges $\{(\mathrm{i}(n_u'), \mathrm{i}(n_w)), (\mathrm{i}(n_w'), \mathrm{i}(n_v'))\}$ from $\mathbb{L}$
49:         Add edges $\{(\mathrm{i}(n_u), \mathrm{i}(n_w)), (\mathrm{i}(n_w'), \mathrm{i}(n_v)), (\mathrm{i}(n_u'), \mathrm{i}(n_v'))\}$ to $\mathbb{L}$
50:     **end if**
51:     $\mathbb{L}_r \leftarrow$ repeated edges in $\mathbb{L}$
52: **end while**
53: **return** $\mathbb{L}$

---

