# OpenReview forum: "Synthetic Data Generation of Many-to-Many Datasets via Random Graph Generation"
_ICLR.cc/2023/Conference — ICLR 2023 poster_

### Official Review · Reviewer_zdm5 · 2022-10-20

**Confidence:** 2
**Correctness:** 3
**Technical Novelty And Significance:** 3
**Empirical Novelty And Significance:** 3
**Recommendation:** 8

**Clarity, Quality, Novelty And Reproducibility:**

Quality:
High.  The paper explains the problem well and uses well-motivated methods to solve the problem.  Furthermore, the paper takes some effort to apply a previous approach to the problem in order to obtain a benchmark.  The evaluation appears reasonable, except the metrics are not well-motivated.

Clarity:
The paper is clearly written.

Originality:
The method solves a novel problem.

**Strength And Weaknesses:**

Strengths
+ Solves a novel problem of generating multiple-table data with rich dependency structure.
+ Does a good job of relating the work to previous approaches and showing the novelty of the approach.

Weaknesses/Questions
- The evaluation criteria are not well explained.  Why does high similarity of mutual information indicate high-quality synthetic data?  Are the metrics used in this paper novel, or have they been previously used in the literature. [after rebuttal: addressed]
- The paper could be improved if some other evaluations founds in the literature were incorporated.  If one were to train a recommender system on the synthetic data, does it find similar features to a recommender system trained on the real data? [after rebuttal: addressed]
- Is there any way to empirically evaluate the quality of the differential privacy in hiding individuals? [after rebuttal: addressed]
- I'm not sure that "faithfulness" is really a property of this method, since it doesn't check the conditional independencies in the real data. However, I think the paper uses the term "faithfulness" to capture the fact that the model doesn't underfit the real data by assuming a constrained dependency structure.  Perhaps a better way to describe this property of the method is that it is potentially consistent for arbitrary dependency structures? [after rebuttal: addressed]

Additional questions/comments after rebuttal:
- How do prove the statement in Appendix C that n_u and n_u' are conditionally independent for sufficiently rich \beta?  Is the proof also in Wagstaff et al. 2019?
- Typo in Appendix D 'Detailed Eexplanation'

Note: I am not familiar with the differential privacy literature, so I defer to other reviewers in the evaluation of the claims of differential privacy.



**Summary Of The Paper:**

Consider the problem of synthetically generating data based on real data with multiple tables and links between tables, such as a movie recommendation dataset, and if possible, in a differentially private way.  This paper suggests a new approach for doing so and applies it to real data.

**Summary Of The Review:**

As the paper convincingly solves a novel and interesting problem, and after rebuttal, explains the evaluation criteria and shows improved results over previous methods, I recommend acceptance.

---

> ### Author Response · Authors · 2022-11-16
> **Author Response to Reviewer zdm5**
>
> We appreciate the reviewer’s time and thoughtful comments. Individual points raised by the reviewer are addressed below.
>
> > The evaluation criteria are not well explained. Why does high similarity of mutual information indicate high-quality synthetic data? Are the metrics used in this paper novel, or have they been previously used in the literature.
> > The paper could be improved if some other evaluations founds in the literature were incorporated.
>
> The pairwise mutual information similarity metric is inspired by the use of pairwise metrics used in [1] where the discrepancy between any pair of marginals is measured by the total variation distance or Cramer’s V with bias correction; our use of mutual information is similar to the latter. The main reason for us to use the (normalized) mutual information is because we hope the score is bounded between 0 and 1 and is higher the better (for easy interpretation).
>
> We agree with the reviewer that an existing metric could be included as the primary metric. Therefore we have adopted the total variation distance based pairwise metric from [1] in the revised draft. Notably the relative order of the metric for the most important target (cross-table) remains unchanged. Please refer to appendix D.2 for the setup and complete results.
>
> > If one were to train a recommender system on the synthetic data, does it find similar features to a recommender system trained on the real data?
>
> Thank you for the suggestion. We added this experiment in appendix D.4. In short, NeuralM2M’s performance is on par with the recommender trained on the real data and confidently overperforming SDV.
>
> > Is there any way to empirically evaluate the quality of the differential privacy in hiding individuals?
>
> Such evaluation is normally done by something called membership inference attacks (MIAs) in the literature. While there are numerous MIAs proposed for classification tasks, it is far more challenging to do so for synthetic data since the output is high dimensional. As far as we are aware, there are only a couple such attacks in a black-box setting [2,3] but they are aimed at a single-table synthetic data (where one row of data corresponds to one individual). It would definitely be very interesting to explore such attacks in a M2M setting (which is a lot more difficult) but we leave that for future work.
>
> > I'm not sure that "faithfulness" is really a property of this method, since it doesn't check the conditional independencies in the real data. However, I think the paper uses the term "faithfulness" to capture the fact that the model doesn't underfit the real data by assuming a constrained dependency structure. Perhaps a better way to describe this property of the method is that it is potentially consistent for arbitrary dependency structures?
>
> Our use of the term “faithfulness” is based on [4] where it is defined as “It is faithful in that it contains sufficient edges to avoid encoding conditional independencies absent from the model.”; we have also defined it accordingly in the beginning of section 4.1. Also note that this faithfulness refers to the novel factorization of U, V, L used by the proposed framework but not individual components. We believe this definition is consistent with what the reviewer suggests as “potentially consistent” and have added a footnote in section 1 to avoid potential confusion.
>
> [1] Yuchao Tao, Ryan McKenna, Michael Hay, Ashwin Machanavajjhala, and Gerome Miklau. Benchmarking differentially private synthetic data generation algorithms. AAAI Workshop on Privacy-Preserving Artificial Intelligence (PPAI), 2022.
>
> [2] Theresa Stadler, Bristena Oprisanu, and Carmela Troncoso. Synthetic Data – Anonymization Groundhog Day. Usenix Security, 2022.
>
> [3] Florimond Houssiau, James Jordon, Samuel N Cohen, Andrew Elliott, James Geddes, Callum Mole, Camila Rangel-Smith, Lukasz Szpruch. PrivE: Empirical Privacy Evaluation of Synthetic Data Generators. NeurIPS 2022 Workshop on Synthetic Data for Empowering ML Research, 2022
>
> [4] Webb, S., Golinski, A., Zinkov, R., Rainforth, T., Teh, Y.W. and Wood, F. Faithful inversion of generative models for effective amortized inference. NeurIPS, 2018

---

> > ### Comment · Reviewer_zdm5 · 2022-11-16
> > **thanks**
> >
> > > Therefore we have adopted the total variation distance based pairwise metric from [1] in the revised draft.
> >
> > Thanks for doing this.  By the way, for someone outside of the synthetic data generation literature, it may help to just add a sentence clarifying that the goal of data generation is to mimic the distribution of the original data, hence why distributional similarity metrics are being used to evaluate the method.  Could you add such a short explanation to the final version?
> >
> > >  In short, NeuralM2M’s performance is on par with the recommender trained on the real data and confidently overperforming SDV.
> >
> > Very nice, thanks for looking into this!
> >
> > > Our use of the term “faithfulness” is based on [4] where it is defined as “It is faithful in that it contains sufficient edges to avoid encoding conditional independencies absent from the model.”; we have also defined it accordingly in the beginning of section 4.1. Also note that this faithfulness refers to the novel factorization of U, V, L used by the proposed framework but not individual components. We believe this definition is consistent with what the reviewer suggests as “potentially consistent” and have added a footnote in section 1 to avoid potential confusion.
> >
> > Thanks.  To further clarify, do other methods fail to be faithful in this sense?

---

> > > ### Author Response · Authors · 2022-11-17
> > > **Follow-up Response**
> > >
> > > We thank the reviewer for the quick response and the follow-up comments.
> > >
> > > > Thanks for doing this. By the way, for someone outside of the synthetic data generation literature, it may help to just add a sentence clarifying that the goal of data generation is to mimic the distribution of the original data, hence why distributional similarity metrics are being used to evaluate the method. Could you add such a short explanation to the final version?
> > >
> > > Thanks for the suggestion. We have added some clarification texts in the “Metrics” under section 5.1 for this.
> > >
> > > > Thanks. To further clarify, do other methods fail to be faithful in this sense?
> > >
> > > Yes. For example, the closest competitive SDV fails to be faithful as it only models one-to-many relationships. Under our notation, one can argue that it uses a factorization of $p(U) p(L_U | U) p(L_V) p(V | U, L_U)$ (where $L_U$ is the $U$ side of the bipartite graph and $p(L_U)$ is fixed such that each node has only one edge). This essentially introduces two independence assumptions:
> > > $L_V$ are $L_U$ are independent, which ignores the node and edge correlations on two sides of the bipartite graph;
> > > $V$ and $U$ are conditionally independent given $L_U$, which ignores the local topology for $V$ in $L$.
> > > Note that these assumptions are “hard-coded” in their modeling approach and cannot be overcome by simply using a more powerful component. This is rooted in the choice of factorization (or the order of how $U, V, L$ are modeled): Under the factorization that SDV takes, in order to achieve faithfulness, one has to come up with a model for $p(L | U)$, which is a non-trivial task as discussed in section 4.1 (the second bullet point).
> > >
> > > Furthermore, when generalizing to more than 2 tables, because SDV can only model one-to-many relationships, it also meets the “multiple parents” issue where it has to choose one parent (at random) for modeling (this has been discussed in the “Generative models for relational data” paragraph in section 3). This is another failure of “faithfulness” for this approach.

---

### Official Review · Reviewer_62xs · 2022-11-01

**Confidence:** 3
**Correctness:** 3
**Technical Novelty And Significance:** 3
**Empirical Novelty And Significance:** 3
**Recommendation:** 6

**Clarity, Quality, Novelty And Reproducibility:**

I think the paper is mostly well-written and polished. There are parts which require knowledge of prior work to understand it better and more details can help here. According to the authors, modeling of tabular data with many to many relationships has been missing from literature and this work tries to address it.

**Strength And Weaknesses:**

Strengths:

* The problem is natural since a lot of real world datasets are tabular and have many to many relationships. A generative model which preserves these properties is important for downstream usecases such as synthetic data generation.
* The paper is well-written; it provides a systematic investigation of approaches to the problem discussing why (extensions of) prior work seem insufficient and how the authors remedy these via proposed approach. Overall, I think the write-up is generally accessible to the wider ICLR audience. There are parts where it gets a little heavy on citing and drawing from prior work (Section 4) without enough details (more on this later).

Weaknesses:

* I think a simple small example of a (made-up) tabular dataset with many-to-many relationships will help the readers. Admittedly, I am not very familiar with this area, and so it took me some time to understand the intended use-case and how the abstraction captures the setting.

* Clarification on modelling assumptions: The proposed approach is to fit a generative model is on a given (single) dataset, which in the simplest setting is a bi-partite graph. In order for the model to learn from this single graph, there need to be (some kind of) sufficient independence between the vertices/edges which provide an overall statistical leverage. The authors do not present a formal statistical model of the underlying graph, however, when fitting the generative model, the factorization implicitly uses certain statistical assumptions (which a-priori, I am not sure if it holds generally); in particular Eqns, (1) and (2).
A formal model would help ascertain the validity of such a decomposition.
Also, a discussion on why these independence assumptions are valid in typical usecases would be helpful.

* The structure of the paper essentially presents a decomposition of the problem into many sub-problems (which is novel, as far as I understand) and then the authors draw from prior work to to solve the sub-problem.
My issue is that the presented description of the technique is minimal and, in places, relegated to a citation. A reader who is not familiar with prior works doesn't really understand the details. Also, it gives the impression that the authors are essentially combining different techniques together.

* (Minor) When discussing various possible factorization of the joint density model, the author argue why modelling of certain factors is not a good idea. While this is good in the sense that it showcases that the authors considered alternatives before finalizing the presented approach, some of the negative aspects are rather "hand-wavy" -- saying that "model can be computational challenge to implement" is perhaps not the strongest defense against using that method.

* A differentially private extension is based on running DP-SGD on the loss obtained after parametrizing the generative model (NueralM2M). The authors say that it has worse performance than the other method. DP-SGD has a clipping parameter and the empirical performance typically is sensitive to its choice. I am wondering if the authors tried to tune this parameter and yet observe this worse performance. Some details would be helpful here.

* Some elaboration om what it means by "faithful" in the context of
"faithful generative models" would be useful.


**Summary Of The Paper:**

The paper consider the problem of synthetic data generation by fitting a generative model to the dataset and then drawing samples from it. The authors consider tabular datasets with many to many relationships.
The key idea that the authors propose is to decompose the likelihood in a certain way and then model each of the factors using appropriate techniques such as sampling structured graphs and constructing graph embeddings. The paper also presents a differentially private extension of the above technique. The authors conduct experiments on a benchmark dataset to adjudge the efficacy of this modeling using appropriate metrics.

**Summary Of The Review:**

The paper presents a systematic investigation of generative modeling of tabular data with many to many relationships, which seems like a natural and interesting problem. The proposed apporach, while natural, requires non-trivial application of prior techniques (such as random graph generation). The apporach is evaluated on a benchmark datasets and the metrics showcase improvements over a prior approach. I am mostly positive about the paper but have some basic questions (see above) which I hope the authors response will adress.

---

> ### Author Response · Authors · 2022-11-16
> **Author Response to Reviewer 62xs (1/2)**
>
> We appreciate the reviewer’s time, valuable feedback, and helpful suggestions. Individual points raised by the reviewer are addressed below.
>
> > I think a simple small example of a (made-up) tabular dataset with many-to-many relationships will help the readers. Admittedly, I am not very familiar with this area, and so it took me some time to understand the intended use-case and how the abstraction captures the setting.
>
> Thanks for the suggestion. We take the main message as the intended use-case of the proposed model is not clear. We’ve included a flow chart (figure 4 in appendix A) to demonstrate how the proposed method is used for synthetic data generation in practice; please also refer to the corresponding updated texts of an example dataset and use cases in appendix A.
>
> > Clarification on modelling assumptions: The proposed approach is to fit a generative model is on a given (single) dataset, which in the simplest setting is a bi-partite graph. In order for the model to learn from this single graph, there need to be (some kind of) sufficient independence between the vertices/edges which provide an overall statistical leverage. The authors do not present a formal statistical model of the underlying graph, however, when fitting the generative model, the factorization implicitly uses certain statistical assumptions (which a-priori, I am not sure if it holds generally); in particular Eqns, (1) and (2). A formal model would help ascertain the validity of such a decomposition. Also, a discussion on why these independence assumptions are valid in typical usecases would be helpful.
>
> We agree with the reviewer that the assumptions in (1) and (2) should be made explicitly.  In short, the assumptions behind (1) is that the representation of local topology from $\beta$ is rich enough such that any two nodes become conditionally independent given the representations of their local topology. Similarly, the assumption behind (2) is that the set embedding from $\gamma$ is rich enough and the choice of deep sets based option does have a theoretic guarantee to make it true with large enough dimension [1]. We have included a more detailed version of this discussion, including a note on fidelity-computation trade-off between different options, in appendix C of the revised draft.
>
> > The structure of the paper essentially presents a decomposition of the problem into many sub-problems (which is novel, as far as I understand) and then the authors draw from prior work to to solve the sub-problem. My issue is that the presented description of the technique is minimal and, in places, relegated to a citation. A reader who is not familiar with prior works doesn't really understand the details.
>
> Thanks for the feedback. We provide some more intuition and descriptions of how learning in prior work is performed. We hope it could bring some more clarity to them for a high-level understanding. However, we note that to obtain a deeper understanding of how each of the components works, we still recommend the readers to consult the original papers as cited.
>
> > Also, it gives the impression that the authors are essentially combining different techniques together.
>
> We would like to emphasize that, while each component is based on existing method, the novelty of the proposed method comes from the novel factorization: In the literature, there is no prior work that consider generating the structure (L) first (without actual node attributes) and use it to generate the rest of the many-to-many data. This new modeling strategy brings random graph generation and popular single-table methods in synthetic data generation together, and opens research opportunities in each individual component.
>
> > "model can be computational challenge to implement"
>
> Modeling p(V | U) implies marginalizing over all possible edges, comparing to p(V | U, L). It is unclear how to do this in practice and also unclear how to use this model once learned such that U and V are correlated with L as the model doesn’t capture any dependencies with L.
>
> ...

---

> > ### Comment · Reviewer_62xs · 2022-11-20
> > **Thanks**
> >
> > I thank the authors for their response. I think the changes made by the authors answers many of my questions and will, in my opinion, help the presentation. About independence assumptions, I understand the informal justification, but I wonder if there is a formal mathematical statement to describe this sense of "rich enough" (while not defining it as the property we finally want). Also, the sentence "this is **more likely** to be true if node embedding (option 2 above) is used" is rather vague. What determines this being likely or not likely? Further, what if they are rich but not "rich enough"? In that case, is there a way to argue about the additional error introduced?
> >
> > Typo in Eqn. 1: The product uses $k$ as the index, but there is no $k$ in the term inside.

---

> > > ### Author Response · Authors · 2022-11-22
> > > **Follow-up Response (1/2)**
> > >
> > > We thank the reviewer for the quick response as well as the follow-up comments and clarifications.
> > >
> > > > About independence assumptions, I understand the informal justification, but I wonder if there is a formal mathematical statement to describe this sense of "rich enough" (while not defining it as the property we finally want).
> > >
> > > To provide a more detailed discussion on the independence assumptions, we start by restating (1) and (2) with intermediate steps (in red) respectively.
> > >
> > > For (1), we have
> > > $$
> > > p(\mathbb{U} \mid \mathbb{L}) = {\color{red} \prod_{i=1}^{\lvert U \rvert} p(n_u^i \mid \mathbb{L}, i)} {\color{red} \approx} \prod_{i=1}^{\lvert U \rvert} p(n_u^i \mid \beta(\mathbb{L}, i)),
> > > $$
> > > where to clarify the potential approximation we replace the original $=$ in (1) by $\approx$; the first equality holds in general. However, it is non-trivial to develop a single-table generative model that could condition on a graph (with arbitrary sizes) and a node (index). Thus we introduce the node embedding $\beta$ to provide a fixed-size embedding for $(\mathbb{L}, i)$.
> > >
> > > As long as $\beta$ gives uniquely identifiable embedding for each $n_u$ in $\mathbb{L}$ with a different local topology, $\approx$ becomes $=$. As an example, in the graph below
> > >
> > > $u_1$ --> ($v_1, v_2$);
> > >
> > > $u_2$ --> ($v_3, v_4$);
> > >
> > > $u_3$ --> ($v_5, v_6$);
> > >
> > > $u_4$ --> ($v_6$)
> > >
> > > - $\beta$ should give the same node embedding for $u_1$ and $u_2$, as their local topology is the same (connected to 2 nodes in $\mathbb{V}$ that do not connect to other nodes in $\mathbb{U}$.
> > > - $\beta$ should give different embeddings for $u_3$ and $u_4$ (which are also different from that of $u_1$ and $u_2$).
> > >
> > > Additionally, we also hope that the embedding from $\beta$ captures the similarity between local topology. In the example above, $\beta$ should give an embedding to $u_3$ that is similar to the embedding of $u_1$/$u_2$ as they both connect to 2 nodes in $\mathbb{V}$ (but those two nodes have a different connection in $\mathbb{V}$). For a similar reason, the embedding for $u_4$ should be more different from that of $u_1$/$u_2$ than that of $u_3$, i.e. $d(h_1, h_4) > d(h_1, h_3)$ ($h_i$ is the node embedding for the node with index $i$) for some distance $d$ (such as L2); a more formal definition could be made based how topology identification is defined in [1]. Note this is not required for the conditional independence assumption but would be beneficial for the performance of the single-table model that conditions on the embedding.
> > >
> > > For (2), we have
> > > $$
> > > p(\mathbb{V} \mid \mathbb{L}, \mathbb{U}) = {\color{red} \prod_{n_v \in \mathbb{V}} p(n_v \mid \mathrm{ne}(n_v))} {\color{red} \approx} \prod_{n_v \in \mathbb{V}} p(n_v \mid \gamma(\mathrm{ne}(n_v)),
> > > $$
> > > where to clarify the potential approximation we replace the original $=$ in (1) by $\approx$; the first equality holds in general. However, it is non-trivial to develop a single-table generative model that could condition on a set with arbitrary number of elements. Thus, we introduce the set embedding $\gamma$ to provide a fixed-size embedding for $\mathrm{ne}(n_v)$ (neighbors of $n_v$).
> > >
> > > For the set embedding $\gamma$, as long as it provides uniquely identifiable embedding for each set, $\approx$ becomes $=$. This is relatively straightforward to justify “how rich this embedding needs to be” based on [2], which states that if the dimension of a representation based on the deep sets architecture is large enough, the model is a universal function representation ($\gamma$ is a function acting on sets).
> > >
> > > Similarly, we also hope $\gamma$ gives similar embeddings to similar sets, which is beneficial for the performance of the single-table model that conditions on the embedding.
> > >
> > > > Also, the sentence "this is more likely to be true if node embedding (option 2 above) is used" is rather vague. What determines this being likely or not likely?
> > >
> > > If the criteria for (1) stated in the previous answer holds, the independence assumption holds for sure. Between option 1 and option 2, option 1 can only get information about $\mathbb{L}$ from nodes within a prescribed depth but option 2 can get information from a wider context. In the case of simple graphs with discontinued subgraphs, option 1 might be enough. However, for arbitrary graphs with unknown structures, we argue it is more likely for option 2 to satisfy the criteria for the independence assumption.
> > >
> > > [1] Giannakis, G.B., Shen, Y. and Karanikolas, G.V., 2018. Topology identification and learning over graphs: Accounting for nonlinearities and dynamics. Proceedings of the IEEE, 106(5), pp.787-807.
> > >
> > > [2] Wagstaff, E., Fuchs, F., Engelcke, M., Posner, I. and Osborne, M.A. On the limitations of representing functions on sets. ICML, 2019

---

> > > ### Author Response · Authors · 2022-11-22
> > > **Follow-up Response (2/2)**
> > >
> > > > Further, what if they are rich but not "rich enough"? In that case, is there a way to argue about the additional error introduced?
> > >
> > > In the case of “rich but not rich enough”, the single-table model will have to learn a mixture model of different node attributes conditioning on the same embedding. Using the example graph above, if $\beta$ gives the same embedding for $u_2, u_3$ but $u_2$ and $u_3$ actually have different distributions, the single-table model needs to learn a mixture of these two distributions. At generation time, the single-table model cannot distinguish between such topology in $\mathbb{L}$ and it can only sample from the mixture at random, hurting the similarity of the synthetic data.
> > >
> > > > Typo in Eqn. 1: The product uses as the index, but there is no  in the term inside.
> > >
> > > Thanks for noting this. I will have it fixed in the final draft.

---

> ### Author Response · Authors · 2022-11-16
> **Author Response to Reviewer 62xs (2/2)**
>
> > DP-SGD has a clipping parameter and the empirical performance typically is sensitive to its choice. I am wondering if the authors tried to tune this parameter and yet observe this worse performance.
>
> We fixed the sensitivity (gradient clipping norm) to 5.0 in our experiments (we missed this detail in our initial draft and have added it to “Privacy budget” in section 5.2). We have tested other values (2.0 and 10.0) and found the performance of NeuralM2M is qualitatively similar to that based on 5.0 for $\epsilon = 0.1, 10.0, 1,000.0$ (i.e. noticeably worse than that of BayesM2M).
>
> Note that the choice of clipping norm, as well as the noise level and total number of iterations given a privacy budget as a whole is a challenging privacy accountant problem for any practical application of DP-SGD. We make our setup as simple as possible (as described in appendix E.2) because the focus of our paper is the many-to-many modeling part. We did not further optimize privacy accountant as it is non-trivial for an optimal tuning both algorithmically and computation-wise.
>
> > Some elaboration of what it means by "faithful" in the context of "faithful generative models" would be useful.
>
> Our use of the term “faithfulness” is based on [2] where it is defined as “It is faithful in that it contains sufficient edges to avoid encoding conditional independencies absent from the model.”; we have also defined it accordingly in the beginning of section 4.1. Also note that this faithfulness refers to the novel factorization of U, V, L used by the proposed framework but not individual components. We believe this definition is consistent with what the reviewer suggests as “potentially consistent” and have added a few clarification sentences in section 1 as well to avoid potential confusion.
>
> [1] Wagstaff, E., Fuchs, F., Engelcke, M., Posner, I. and Osborne, M.A. On the limitations of representing functions on sets. ICML, 2019
>
> [2] Webb, S., Golinski, A., Zinkov, R., Rainforth, T., Teh, Y.W. and Wood, F. Faithful inversion of generative models for effective amortized inference. NeurIPS, 2018

---

### Official Review · Reviewer_Ysb4 · 2022-11-02

**Confidence:** 2
**Correctness:** 3
**Technical Novelty And Significance:** 2
**Empirical Novelty And Significance:** 2
**Recommendation:** 6

**Clarity, Quality, Novelty And Reproducibility:**

The paper is well-written and easy to follow. As mentioned above, there are some statements in the paper that need more clarification. I have some concerns about the novelty since the privatization part is quite straightforward and there is not any theoretical result to provide any privacy or utility guarantee.

**Strength And Weaknesses:**

Strength:

Developing differentially private SDG is an important and well-motivated topic. The idea to decompose the joint distribution into multiple components and treat each component separately is interesting.

Weakness:

1. The privatization step in section 4.3 is straightforward and may not be scalable. It looks like in the case of prefix embeddings, there are no learnable parameters for $P(U|L)$ and $P(V|L, U)$, and the privatization process reduces into a direct application of Laplace mechanism to the BJD matrix. Moreover, the noise added to each entry of the BJD matrix scales with the number of nodes, which can be prohibitively large for complicated graphs with plenty of nodes.  Finally, the authors mention that we can use DP-SGD to train the embedding. If that is the case, it would be better if the authors can present the training objective and algorithm details in the paper.
2. The descriptions of the distribution model in section 4.2 are not clear. In particular, the authors only provide very general introduction to different node embeddings β or aggregators γ. It would be better if the authors can present more details about the how these embeddings and aggregators work.
3. In the experiment, the authors mention that “we use node2vec to extract node embeddings”. I am wondering how the authors privatize this embedding learning step to prevent the privacy leakage.

Question:
1. This is a follow-up question about the privatization step in experiments. In section 4.3, the authors mention that "otherwise we use DP-SGD to train the embedding". I notice that the author do not mention any training details concerning DP-SGD in experiment (section 5), so my question is that whether the authors implement DP-SGD in any of the experiments. If yes, it would be great if the authors can provide more relevant implementation details.



**Summary Of The Paper:**

This paper presents an algorithm of synthetic data generation for many-to-many dataset. In particular, the authors decompose the overall distribution into three components and model the relationship in each component independently. Furthermore, the authors also present a private version of the algorithm and conduct experiments on real-world datasets.

**Summary Of The Review:**

The paper presents a synthetic data generation algorithm for many-to-many datasets. The idea is well-motivated and the authors do a good job in presenting the algorithms. However, I am not sure the novelty is enough since the privatization part in sec 4.3 is a direct application of Laplace mechanism for pre-fixed embedding and the authors only briefly mention about training embedding privately with DP-SGD without providing any further details. Therefore, the paper can be improved if the authors can
1. Add more details about the embedding learning with DP-SGD
2. Provide theoretical guarantee (privacy or utility) of the proposed algorithm.

---

> ### Author Response · Authors · 2022-11-16
> **Author Response to Reviewer Ysb4 (1/2)**
>
> We appreciate the reviewer’s time and thoughtful comments/questions. Individual points raised by the reviewer are addressed below.
>
> > The privatization step in section 4.3 is straightforward and may not be scalable. It looks like in the case of prefix embeddings, there are no learnable parameters for P(U|L) and P(V|L,U), and the privatization process reduces into a direct application of Laplace mechanism to the BJD matrix. Moreover, the noise added to each entry of the BJD matrix scales with the number of nodes, which can be prohibitively large for complicated graphs with plenty of nodes.
>
> First, we would like to emphasize that the main focus and contribution of this paper is the M2M modeling (newly proposed flexible end-to-end framework, novel factorization model, pros/cons discussion of individual components, 2 specific initializations of our framework which both beat alternatives (i.e., SDV)). As a secondary contribution, we extend the framework to be DP but  decide to rely on existing privacy techniques. We believe that there would be more optimal ways to achieve DP but we leave that for future work.
>
> Second, we would like to point out that, for the graph generation part, it is the largest connectivity rather than the number of nodes that determines the noise added; please see the second paragraph in section 4.3, based on [1].
>
>
> > Finally, the authors mention that we can use DP-SGD to train the embedding. If that is the case, it would be better if the authors can present the training objective and algorithm details in the paper.
> > In the experiment, the authors mention that “we use node2vec to extract node embeddings”. I am wondering how the authors privatize this embedding learning step to prevent the privacy leakage.
> > This is a follow-up question about the privatization step in experiments. In section 4.3, the authors mention that "otherwise we use DP-SGD to train the embedding". I notice that the author do not mention any training details concerning DP-SGD in experiment (section 5), so my question is that whether the authors implement DP-SGD in any of the experiments. If yes, it would be great if the authors can provide more relevant implementation details.
>
> The only place where DP-SGD is used in our experiments is for NeuralM2M where both the set embedding and the conditional generative models are learned. The set embedding (based on deep sets) is learned end-to-end together with the generative models using their own objectives. As the models are normally learned using SGD, to make them DP, we simply replace the optimizer by DP-SGD. We rely on a standard implementation of DP-SGD for which we set the sensitivity (gradient clipping norm) as 5.0 and compute the corresponding noise level given a privacy budget evenly distributed among a prescribed number of iterations. Please refer to “Privacy budgets” in section 5.2 and “Privacy accountant” in appendix E.2 for details.
>
> For node2vec, we would like to clarify that there is no learning parameter from this approach (i.e. no parameters computed from the real data are used at sampling time for $\beta$ ). At sampling time, the only input to node2vec is the DP graph thus there is no privacy leakage as DP is robust to post-processing. We apologize for some of the misleading descriptions in the original draft and we have had them corrected in the revised draft.
>
> As per the other question, we have included a more detailed description of how learning for $\beta$ and $\gamma$ works in the “Learning” paragraphs in section 4.2.2 and section 4.2.3.
>
> > The descriptions of the distribution model in section 4.2 are not clear. In particular, the authors only provide very general introduction to different node embeddings β or aggregators γ. It would be better if the authors can present more details about the how these embeddings and aggregators work.
>
> The options for the node embeddings $\beta$ and the aggregators $\gamma$ in section 4.2 are either parameter-free options or learned approaches based on existing methods. For the former, we have provided a detailed description of how they are implemented when used in our experiments in the paragraph titled “Two model instances from the proposed framework” in section 5.1 as well as appendix E.2. For the latter, we have included a more detailed description in the “Learning” paragraphs in section 4.2.2 and section 4.2.3.
>
> We would like to emphasize that, while each component is based on an existing method, the novelty of the proposed method comes from the novel factorization. To the best of our knowledge in the literature, there is no prior work that considers generating the structure (L) first (without actual node attributes), and then using it to generate the rest of the many-to-many data. This new modeling strategy brings random graph generation and popular single-table methods in synthetic data generation together, and opens research opportunities in each individual component.
>
> ...

---

> ### Author Response · Authors · 2022-11-16
> **Author Response to Reviewer Ysb4 (2/2)**
>
> > I have some concerns about the novelty since the privatization part is quite straightforward and there is not any theoretical result to provide any privacy or utility guarantee.
>
> We would like to re-emphasize that our main focus is M2M modeling so we decide to rely on existing privacy techniques. We believe that there would be better ways to achieve DP in this context but we leave that for future work.
>
> In terms of theoretical results of privacy guarantee, our definition 1 together with the privatization steps described in sec 4.3 indeed provide theoretical guarantee: even though the privatization part is straightforward, it does have theoretical implications (we keep the nodes private) based on the DP composition theorem.
>
> Finally, in the context of DP synthetic data, while DP mechanisms provide privacy guarantees, assuring utility is very challenging (since generally DP mechanisms rely on randomness/perturbation) and is still an open problem. This motivates further empirical evaluation work as claimed by [2, 3].
>
> [1] Davide Proserpio, Sharon Goldberg, and Frank McSherry. A workflow for differentially-private graph synthesis. In Proceedings of the 2012 ACM workshop on Workshop on online social networks, 2012.
>
> [2] Michael Hay, Ashwin Machanavajjhala, Gerome Miklau, Yan Chen, and Dan Zhang. Principled evaluation of differentially private algorithms using dpbench. In SIGMOD, 2016.
>
> [3] Yuchao Tao, Ryan McKenna, Michael Hay, Ashwin Machanavajjhala, and Gerome Miklau. Benchmarking differentially private synthetic data generation algorithms. AAAI Workshop on Privacy-Preserving Artificial Intelligence (PPAI), 2022.

---

> > ### Comment · Reviewer_Ysb4 · 2022-11-16
> > **More clarification on the application of Laplace mechanism**
> >
> > Thanks for the detailed response from the authors. It has been very helpful. However, I still have some confusions in the privatization step. From [1], given any function (algorithm) $f:\mathbb{N}^{|\mathcal{X}|}\to \mathbb{R}^k$, the Laplace mechanism is defined as
> > $$M(x, f, \epsilon) = f(x) + (Y_1,\dots Y_k)$$
> > where $Y_i$ are i.i.d r.v from $\text{Lap}(\Delta f/\epsilon)$ and $\Delta f$ is the $l_1$ sensitivity of $f$.
> >
> > In other words, the Laplace mechanism is realized by adding *same amount* of Laplace noise to *every* entry of the output, and the noise scales with the $l_1$ sensitivity of the computation function $f$.
> >
> > However, in section 4.3, the authors try to achieve $\epsilon$-DP by adding noise sampled from $\text{Lap}(S_{i, j}/\epsilon)$ to $i_{th}$ row and $j_{th}$ column entry of the BJD matrix. It looks like the authors add different amounts of noise to different entries of output. If that is the case, this is deviated from the standard Laplace mechanism and I am wondering what mechanism or result the authors rely on to conclude the proposed method in section 4.3 is $\epsilon$-DP.
> >
> > I understand that the DP-part is secondary and utility guarantee is hard to get. But it would be great if the authors can give a formal privacy guarantee written in the form of a theorem including a clear definition of the data universe, the analysis of $l_1$ sensitivity (if needed) and, most importantly, why the claimed privacy guarantee can hold.
> >
> > Thanks!
> >
> > [1] Dwork, Cynthia, and Aaron Roth. "The algorithmic foundations of differential privacy." Foundations and Trends® in Theoretical Computer Science 9.3–4 (2014): 211-407.

---

> > > ### Author Response · Authors · 2022-11-17
> > > **Follow-up Response**
> > >
> > > We thank the reviewer for the quick response as well as the follow-up comments and clarifications.
> > >
> > > > It looks like the authors add different amounts of noise to different entries of output. If that is the case, this is deviated from the standard Laplace mechanism and I am wondering what mechanism or result the authors rely on to conclude the proposed method in section 4.3 is $\epsilon$-DP.
> > >
> > > We agree with the reviewer that the standard definition of the Laplace mechanism applies the same amount of noise (i.e., drawn from the Laplace distribution with the same standard deviation because the sensitivity is constant) to every entry of the output.
> > >
> > > We, however, do not use the standard Laplace mechanism. As mentioned before, in our paper we use already existing DP mechanisms to guarantee privacy. Specifically, for $p_{\theta_1}(\mathbb{L})$ in the second paragraph of 4.3, we rely on the non-uniform scale used by [4]. For context, in section 1.1 of [4], the authors claim that "for each (d1, d2) entry of the JDD, it suffices to add noise proportional to 4 max(d1, d2)." In turn, they rely on the results from [3] to argue that "the noise required to protect the privacy of the JDD can be non-uniform." We hope that this clarifies the soundness of the non-uniform noise we are using.
> > >
> > > > it would be great if the authors can give a formal privacy guarantee written in the form of a theorem including a clear definition of the data universe, the analysis of $l_{1}$ sensitivity (if needed) and, most importantly, why the claimed privacy guarantee can hold.
> > >
> > > We would like to reiterate that in Definition 1, we provide a definition of the DP process and what we mean by neighboring datasets (i.e., our “unit” of privacy) – a single node in either $\mathbb{L}$ or $\mathbb{U}$. We added a sentence to define the data universe $\mathcal{B}$ as a dataset with many-to-many relationships.
> > >
> > > Furthermore, given our factorization model, we spend 4.3 explaining how we guarantee DP of the 3 sub-models (relying on existing DP mechanisms). By sequential composition [1,2] this makes the overall model DP. We added Theorem 1 in 4.3 to summarize this.
> > >
> > > [1] Dwork, Cynthia, Frank McSherry, Kobbi Nissim, and Adam Smith. "Calibrating noise to sensitivity in private data analysis." In Theory of cryptography conference, 2006.
> > >
> > > [2] Dwork, Cynthia, Krishnaram Kenthapadi, Frank McSherry, Ilya Mironov, and Moni Naor. "Our data, ourselves: Privacy via distributed noise generation." In Annual international conference on the theory and applications of cryptographic techniques, 2006.
> > >
> > > [3] Alessandra Sala, Xiaohan Zhao, Christo Wilson, Haitao Zheng, and Ben Y Zhao. Sharing graphs using differentially private graph models. In Proceedings of the 2011 ACM SIGCOMM conference on Internet measurement conference, 2011.
> > >
> > > [4] Davide Proserpio, Sharon Goldberg, and Frank McSherry. A workflow for differentially-private graph synthesis. In Proceedings of the 2012 ACM workshop on Workshop on online social networks, 2012.

---

> > > > ### Comment · Reviewer_Ysb4 · 2022-11-17
> > > > **Follow up**
> > > >
> > > > Thanks for the clarification. I raised the score based on the discussion.

---

> > > > > ### Author Response · Authors · 2022-11-19
> > > > > **Thank You**
> > > > >
> > > > > We thank the reviewer for their feedback, improving our work, and raising their score.

---

### Author Response · Authors · 2022-11-16
**Revised Draft Uploaded**

We sincerely appreciate the reviewers’ time and helpful feedback. We have updated our paper with new content (discussions, clarifications, examples, results) suggested by the reviewers, marked in red. Individual points raised by each reviewer are addressed in separate comments.

---

### Author Response · Authors · 2022-11-19
**Summary for Discussion Stage 1**

We thank all reviewers for engaging in the process in good faith as well as for their valuable time, constructive feedback, and intriguing questions during the first stage of the discussion. We believe we have addressed all of the reviewers’ comments either in the revised draft, or in the author responses posted separately to each reviewer. We hope the reviewers’ original concerns have been addressed and we believe their incorporated feedback significantly improved the quality of our work. We would really appreciate it if the reviewers could update their scores if they find the manuscript improved.

Below is a summary of changes we made to the manuscript during the first stage of discussion:

1. Illustration of how generative models are used for SDG (found in appendix A; in response to reviewer `62xs`).
2. Clarification on the term “faithfulness” (found in footnote 1; in response to reviewers `62xs` and `zdm5`).
3. Addition of the data universe definition in Definition 1 (found in section 2; in response to reviewer `Ysb4`).
4. Explanation of how node embedding $\beta$ and set embedding $\gamma$ work and are learned (found in section 4.2.2 and 4.2.3 in response to reviewers `Ysb4` and `62xs`).
5. Clarification on independence assumptions behind equation 1 and equation 2 (found in appendix C; in response to reviewers `62xs` and `zdm5`).
6. Clarification on DP modeling of $p_{\theta_{2}}(\mathbb{U}\mid\mathbb{L})$ (found in section 4.3; in response to reviewer `Ysb4`).
7. Introduction of Theorem 1 (found in section 4.3; in response to reviewer `Ysb4`).
8. Clarification on goal of metrics (found in section 5.1; in response to reviewer `zdm5`).
9. Introduction of more experiments, results, and analysis:
    * Total variation similarity (found in appendix D.3; in response to reviewer `zdm5`)
    * Recommender system (found in appendix D.4; in response to reviewer `zdm5`)
10. Clarification on the used clipping values in DP-SGD (found in section 5.2; in response to reviewer `62xs`).

---

### Public Comment · ~Carl_Yang1 · 2023-03-23
**Great work and a missing reference**

Dear authors,

This is an interesting paper and we enjoyed reading it. We had a pretty relevant work before and hope you could consider properly referencing it in a later version of your paper or future related works. Thanks.

Yang, Carl, et al. "Secure deep graph generation with link differential privacy." IJCAI (2021).

---

### Decision · Program_Chairs · 2023-01-20

**Decision:**

Accept: poster

**Justification For Why Not Higher Score:**

The contributions are above the bar of acceptance but the level of impact is not very high to merit an oral or spotlight slot.

**Justification For Why Not Lower Score:**

The paper makes enough contributions to merit acceptance.

**Metareview: Summary, Strengths And Weaknesses:**

The paper studies synthetic data generation for tabular datasets with many-to-many relationships among them. The main novelty in this paper is modeling the many-to-many relationships via a new factorization model and proposing a flexible end-to-end framework for synthetic generation based on this new model. The framework is also extended to ensure a formal privacy guarantee via adopting the notion of approximate differential privacy.

All reviewers agreed that the contributions are strong enough to merit acceptance and will be useful to the community, especially given the importance and the practical significance of the problem of synthetic data generation. The authors have been very responsive to the reviewers' initial concerns and made necessary adjustments and clarifications to improve the readability of their paper and to emphasize their contributions.

**Note From Pc:**

if the above contains the word "oral" or "spotlight" please see: "oral" presentation means -> notable-top-5% and "spotlight" means -> notable-top-25%. As stated in our emails, we are disassociating presentation type from AC recommendations